# Scalable Ensemble Federated Learning with Enhanced Open-Set Recognition

**Mustafa Siddiqui**                                   *mustafa.siddiqui@lums.edu.pk*
*Department of Electrical Engineering*
*Lahore University of Management Sciences*

**Muhammad Tahir**                                          *tahir@lums.edu.pk*
*Department of Electrical Engineering*
*Lahore University of Management Sciences*

**Reviewed on OpenReview:** *https://openreview.net/forum?id=QnnCYOfuUI*

## Abstract

Consensus-driven parameter averaging constitutes the dominant paradigm in federated learning. Although many methods incorporate auxiliary mechanisms or refinements, repeated round averaging remains their fundamental backbone. This paradigm inherently depends on repeated rounds of client–server communication to maintain consensus. The reliance on repeated communication is further amplified in regimes with high data heterogeneity and large client populations, as shown across numerous studies. This behavior arises from optimization drift in out-of-distribution settings, where client objectives differ and multi-step local SGD updates increasingly diverge, making consensus difficult to maintain. We argue that an emerging alternative, ensemble with abstention, provides a more suitable framework for addressing these issues. Rather than enforcing consensus across diverging client objectives, this approach constructs a specialized mixture-of-experts model by preserving client-specific models and selectively aggregating their predictions. As a one-shot FL method, it eliminates the need for repeated communication rounds altogether. Moreover, supported by both theoretical and empirical analysis, we show that this paradigm sidesteps cross-client drift and is inherently less sensitive to data heterogeneity. Despite these advantages, ensemble with abstention introduces two fundamental challenges. First, its performance depends on the design of the open-set recognition (OSR) task, which directly affects performance under heterogeneity. Second, and more critically, preserving client-specific models causes linear growth in model size with the number of clients, limiting scalability. As a step toward addressing these limitations, we introduce FedSOV, which incorporates improved negative sample generation to prevent shortcut cues in the OSR task and employs pruning to address the scalability problem. We show that pruning provides a practical and effective solution to the scalability problem while simultaneously enhancing generalization, yielding higher test accuracy. Across datasets, our method demonstrates clear improvements in highly heterogeneous regimes compared to both the ensemble baseline FedOV and the strongest parameter-averaging method, FedGF. **Code is available at:** https://github.com/Mustafa00124/FedSOV

## 1 Introduction

Real-world distributed machine learning scenarios often involve strict privacy constraints, where sharing raw data between parties is not permitted. Federated Learning (FL) has emerged as a popular paradigm in such settings, enabling clients to collaboratively train a global model without exchanging their private data Yang et al. (2019). A key objective in FL is to learn a model that generalizes well across all client distributions while keeping the confidentiality of individual data intact.

Standard FL framework, FedAvg, is based on parameter averaging where clients perform local training before sending it to a central server for averaging over multiple communication rounds to produce a global model McMahan et al. (2017). While simple and widely adopted, FedAvg relies on the assumption that client data is independent and identically distributed (IID), an assumption that rarely holds in practice. In reality, federated systems often involve statistical heterogeneity, where clients have data drawn from different distributions. By averaging parameters, FedAvg seeks consensus across clients even when their local objectives diverge, making parameter averaging unreliable and slow in convergence, requiring massive communication rounds. Additional challenges arise from system heterogeneity, where clients differ in compute power and availability; model heterogeneity, where clients may use different architectures; and continual learning, where clients receive new data over time Pei et al. (2024); Criado et al. (2022).

Recently ensemble-based approaches have been proposed to address the communication efficiency and heterogeneity problems. While earlier works showed that ensemble methods perform well under homogeneous data and primarily focused on improving communication efficiency, more recent works have demonstrated their effectiveness to heterogeneous scenarios Diao et al. (2023). The state-of-the-art ensemble method, FedOV, uses open-set recognition (OSR) to identify an introduced unknown class while retaining the discriminative power of local models. By stacking all local models and allowing them to abstain on inputs outside their expertise, this mechanism naturally avoids issues such as parameter misalignment Wang et al. (2020), and is inherently more robust to statistical, system, and model heterogeneity. Notably, the performance of this approach hinges on how effectively the OSR mechanism handles out of distribution shift at the local level. Although this approach provides a way to address heterogeneity in ensemble methods, the primary limitation is that ensemble size grows linearly with the number of clients, making this approach impractical at scale.

In this paper, we argue that parameter averaging is highly sensitive to heterogeneity, leading to unavoidable optimization error inherent to the aggregation mechanism and, therefore, requiring substantially more communication rounds to achieve acceptable performance under high heterogeneity. Instead, we theoretically and empirically show that an ensemble with abstention is better suited for aggregation in heterogeneous scenarios. The key idea is that when clients possess disjoint information, a specialization step within the solution is required to preserve each client's unique local knowledge. Because it is a type of One-Shot FL method, this approach avoids communication costs entirely. However, the main limitation is that it leads to increased model size, a trade-off that we show can be managed through proper pruning techniques. Building on this insight, we introduce **FedSOV**: Federated Scalable Open Voting. Our main contributions in this paper are:

- We analyze and compare the optimization error of ensemble-based aggregation (with abstention) and parameter averaging on the global FL objective, providing a paradigm-level theoretical perspective on the trade-offs and limitations of the two approaches.

- We introduce FedSOV, a scalable ensemble-based method that mitigates the OSR bottleneck and resolves the inherent scalability issues of prior ensemble approaches through principled pruning. FedSOV achieves state-of-the-art performance against recent ensemble and parameter-averaging methods tailored for heterogeneous FL, particularly in high-heterogeneity settings, while maintaining the same parameter count as parameter averaging methods.

The remainder of the paper is organized as follows. In Section 2, we review related work in FL. Section 3 presents a comparative analysis of parameter averaging and ensemble-based methods. In Section 4, we introduce our proposed method, FedSOV. Section 5 covers our experimental setup/results, and we conclude in Section 6.

## 2 Background and Related Work

**Early Approaches to Non-IID Federated Learning:** FL must confront data heterogeneity across clients, which significantly degrades its performance. The seminal FedAvg algorithm McMahan et al. (2017) performs well under IID data, but its accuracy degrades under non-IID settings. When clients have divergent data distributions (e.g., different label proportions), the global model update from averaging local parameters

can diverge from the true descent direction. Numerous works have documented this issue: for example, Zhao et al. (2018) showed that highly skewed label distributions can cause FedAvg's accuracy to drop by over 50%, and Li et al. (2020) introduced FedProx to stabilize training via a proximal term. Even under IID data, averaging neural network weights can suffer from permutation inconsistency, leading to misaligned layers as noted by FedMA Wang et al. (2020). Mitigation strategies include correction of local updates (SCAFFOLD) Karimireddy et al. (2020), gradient harmonization Zhang et al. (2023), promoting flatter minima Qu et al. (2022a), explicit local-global alignment Li et al. (2021), and data sharing/augmentation.

**Recent Approaches to Label Skew in Federated Learning**: Despite these advances, there is still no single clear solution to the label skew problem, and a variety of techniques continue to be proposed. FedConcat Diao et al. (2024) clusters clients according to their label distributions, trains cluster-specific models via FedAvg, and constructs a global model by concatenating feature extractors across clusters while averaging only the classifier head. FedVLS Guo et al. (2025) addresses vacant-class scenarios by combining vacant-class distillation with logit suppression for non-local classes, thereby improving recognition of unseen labels while retaining parameter averaging. FedLEC Yu et al. (2025) also aims to mitigate the bias due to missing labels in federated learning of spiking neural networks through label-balanced local training and cross-client distillation. In addition, other approaches reflect different directions: FedLMD Lu et al. (2023) employs label-masking distillation to enhance minority-class learning, while FLea Xia et al. (2024) introduces obfuscated feature sharing with mixup-based augmentation under FedAvg. A particularly promising line of research focuses on sharpness-aware optimization, first explored in federated settings by FedSAM Qu et al. (2022a). Building on this idea, MoFedSAM Qu et al. (2022b) and the recent FedGF Lee & Yoon (2024) pursue flatter minima to alleviate client-drift and reduce the risk of model collapse under disjoint data.

**Ensemble-Based Approaches in Federated Learning**: Ensemble methods in FL were originally introduced to address the communication bottleneck, particularly in one-shot settings where each client trains locally and sends a model to the server only once Guha et al. (2019). Early designs simply averaged client models in a single round (one-shot FedAvg), but this often yielded suboptimal results even in IID scenario. This led to the alternative of combining outputs rather than weights, forming an ensemble at the server. While naive voting or averaging of predictions works for IID data, it fails in label-skewed settings, as models tend to misclassify unseen classes into seen ones, causing majority voting to collapse. Methods such as FEDBE Chen & Chao (2020), which treats global aggregation as a Bayesian ensemble over multiple global models, and FEDBOOST Hamer et al. (2020), which builds ensembles via weighted model averaging with theoretical guarantees for certain distributions, extended the ensemble concept but still faced this limitation. FEDOV Diao et al. (2023) addressed the problem by equipping each model with an open-set recognition mechanism that trains with synthetic outlier samples labeled as an unknown class, enabling models to abstain on unfamiliar inputs and improving ensemble decisions under heterogeneity. This OSR-based ensemble showed strong potential but has remained relatively underexplored. Another class of one-shot federated learning methods leverages generative models to approximate the global data distribution and synthesize data at the server Yang et al. (2024). However, these approaches face a fundamental tradeoff between privacy and performance. To mitigate this, ensemble-based methods use client models as teachers to guide synthetic data generation via data-free distillation, with approaches such as Co-Boosting Dai et al. (2024) further improving both the ensemble and synthetic data through iterative refinement. A separate line of work focuses on model fusion, where methods like FuseFL Tang et al. (2024) explicitly address client heterogeneity by combining models in a layer-wise manner with adapter modules. In contrast, hybrid approaches such as FENS Allouah et al. (2024) and FedConcat Diao et al. (2024) combine parameter averaging and ensembling to balance specialization and aggregation under heterogeneous data. Most of these methods typically focus on reducing communication and accept fewer rounds as a tradeoff for lower performance, which is expected to be significant under heterogeneous data. In contrast, we build on the FedOV framework, arguing that ensemble-based approaches augmented with OSR provide a more effective mechanism for handling heterogeneity than parameter averaging, while retaining one-shot efficiency.

**Model Compression in Ensemble FL**: Although ensemble-based approaches were initially valued for reducing communication cost in federated learning and have recently shown strong potential in addressing heterogeneity, they carry a critical caveat: scalability. The scalability problem in FL has been recognized since early work such as Guha et al. (2019), where the cost of communicating and aggregating full models

was shown to be a major bottleneck. Even if parameter averaging is avoided, a practical challenge for ensemble-based FL is the rapid growth in model size and deployment cost as the number of clients increases. Unlike FedAvg's single global model, an ensemble that retains all local models can become prohibitively large, with total parameters scaling linearly with the number of clients. This scalability issue makes vanilla ensembling impractical in large networks or on edge devices. A common strategy to address this has been knowledge distillation, explored since FedMD Li & Wang (2019), which demonstrated that heterogeneous models can collaborate via public-data logit sharing without revealing architectures or private data. In the ensemble compression setting, the server uses a public or generated dataset to train a compact global model that imitates the ensemble's predictions. FedDF Lin et al. (2020) and related approaches exemplify this strategy, but they often assume access to auxiliary data at the server. An alternative line of work explores model pruning to compress federated models. Li et al. Li et al. (2024) propose a client-side pruning approach where each client trains a local model and prunes less significant parameters before sending to the server, which then aggregates these slimmed models. Building on a similar intuition, our work (FedSOV) applies pruning in the context of ensembles.

## 3 Parameter Averaging vs. Ensemble with Abstention

We consider an FL setting with $C$ clients indexed by $c \in \{1, \ldots, C\}$. Data are generated from a mixture of client-specific distributions: let $C$ also denote the (random) client identity, and let $(X, Y)$ be the random input–label pair. The global data distribution is modeled as

$$P(X, Y) = \sum_{c=1}^{C} P(C = c) \, P(X, Y \mid C = c), \tag{1}$$

where $\pi_c := P(C = c)$ is the mixture weight of client $c$ and $P_c(X, Y) := P(X, Y \mid C = c)$ is the client-$c$ local distribution. A predictor $f(\cdot; \theta)$ with parameters $\theta$ maps inputs $X$ to predictions $\hat{y} = f(X; \theta)$, and performance is measured by a per-sample loss $\ell(\hat{y}, y)$. The global objective is the expected loss under the mixture distribution $L(\theta) = \mathbb{E}_{(X,Y) \sim P}\big[\ell(f(X; \theta), Y)\big]$ which can be written equivalently as a weighted sum of client risks:

$$L(\theta) = \sum_{c=1}^{C} \pi_c \, \mathbb{E}_{(X,Y) \sim P_c}\big[\ell(f(X; \theta), Y)\big]. \tag{2}$$

Defining the client-specific objective $L_c(\theta) := \mathbb{E}_{(X,Y) \sim P_c}\big[\ell(f(X; \theta), Y)\big]$ the FL goal is to find $\theta$ that minimizes $L(\theta)$ while each client can only estimate and optimize its own $L_c(\theta)$ from local data.

There are two natural ways to approach this problem. The first operates in parameter space. Clients take gradient descent steps on their local model parameters, and the server averages all the clients' model parameters. This aggregation can be interpreted as an approximate descent step on the global objective, and this is essentially the mechanism underlying methods such as FedAvg. The second approach operates in function space. Instead of averaging parameters, clients return their local predictors, and the server combines their output by creating an ensemble. Under suitable assumptions, this ensemble can be interpreted as an approximate closed-form solution to the global objective.

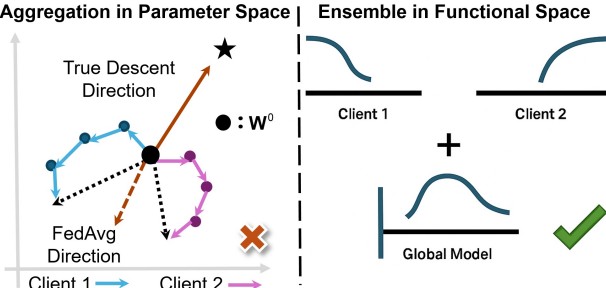

Figure 1: **Left:** Parameter-space averaging can deviate significantly from the true descent direction, leading to unbounded error. **Right:** Functional-space aggregation with abstention preserves each client's specialization, enabling robust stitching of functions into a globally consistent model. Aggregation error here depends primarily on OSR performance.

To understand how these two approaches compare, it is useful to explicitly characterize the optimization error introduced by each aggregation mechanism when

optimizing the global objective. We do so by decomposing the overall risk into terms that isolate the error induced by aggregation. Under a simple setup, the error resulting from a single round of parameter averaging can be quantified in terms of the level of client heterogeneity, the number of local epochs, and the learning rate. In contrast, the error of the ensemble method can be expressed in terms of the oracle selection error and client disagreement, which captures the degree of heterogeneity among clients. Presenting the errors in this form clarifies how heterogeneity affects the two approaches in fundamentally different ways, facilitates direct comparison, and highlights the specific bottlenecks that limit their performance. This perspective motivates the modifications we introduce to make the ensemble approach more competitive with state-of-the-art federated learning methods. Our theoretical analysis should be interpreted as a conceptual comparison between parameter averaging drift and expert-selection error. The analysis does not prescribe how experts should be selected, nor does it derive the proposed smoothed shuffled negative generation strategy. The proposed OSR design is empirically motivated and evaluated experimentally. The proofs of both theorems are in Appendix A.

## 3.1 Optimization Error

We first analyze aggregation in parameter space and present a theorem that characterizes the excess risk over the centralized optimization, introduced by a single round of parameter averaging when optimizing the global federated learning objective.

**Assumption** (Regularity Conditions)**.** *The following conditions hold:*

*(i)* **Smoothness.** *Each local objective $L_c$ is $\beta$-smooth, i.e.,*

$$\|\nabla L_c(\theta) - \nabla L_c(\theta')\| \leq \beta \|\theta - \theta'\|.$$

*(ii)* **Bounded Gradient Dissimilarity.** *For all $\theta$ and all clients $c$,*

$$\|\nabla L_c(\theta) - \nabla L(\theta)\| \leq B_c.$$

*(iii)* **Lipschitz Global Loss.** *The global loss $L(\cdot)$ is $G$-Lipschitz, i.e.,*

$$|L(\theta) - L(\theta')| \leq G \|\theta - \theta'\|.$$

**Theorem 1** (One-Round Parameter Averaging Optimization Error)**.** *Under the regularity conditions stated above, the global loss of the averaged model after $T$ local gradient descent steps satisfies, for $T > 0$,*

$$\mathcal{E}_{\text{avg}} \leq \underbrace{\mathbb{E}_{(x,y)\sim\mathcal{D}}\big[\ell\big(f_{\tilde{\theta}^{(T)}}(x), y\big)\big]}_{\textit{centralized error}} + \underbrace{\frac{G\bar{B}}{\beta}\Big((1+\eta\beta)^T - 1\Big)}_{\textit{parameter averaging drift}}.$$

*Here, $G$ denotes the Lipschitz constant of the global loss, $\beta$ is the smoothness constant of the local objectives, $\eta$ is the local learning rate, and $\bar{B} = \frac{1}{C}\sum_{c=1}^{C} B_c$ is the average gradient dissimilarity across clients.*

**Interpretation.** The bound shows that the optimization error depends both on how well the corresponding centralized optimization trajectory performs and on an additional error introduced by parameter aggregation. Even when the centralized objective is well optimized, parameter averaging incurs an extra drift term that grows exponentially with the number of local update steps $T$ and depends on the geometric properties of the loss landscape, such as its curvature (captured by $\beta$) and sensitivity (captured by $G$), as well as on client heterogeneity through $\bar{B}$. Intuitively, when local objectives deviate substantially from the global objective, enforcing consensus through parameter averaging becomes increasingly misaligned. Importantly, when $T = 1$, this aggregation error vanishes. As discussed in the remark in subsec:theorem-1, due to the linearity of gradients, a single local update followed by averaging exactly recovers a gradient descent step on the global objective, explaining why FedSGD performs exact optimization of the global loss. However, for $T > 1$, the accumulation of aggregation error can significantly slow convergence, requiring many communication rounds to compensate, and may even lead to oscillatory behavior around suboptimal solutions.

Next, we present a theorem that characterizes the excess risk of the learned ensemble relative to the oracle ensemble.

**Theorem 2** (Optimization Error of the Learned Ensemble). *Under the definitions and assumptions stated in Appendix A.2, the error of the learned ensemble predictor satisfies*

$$\mathcal{E}_{\text{ens}} \; \leq \; \underbrace{\mathbb{E}_X\big[\ell\big(\bar{f}(X), y\big)\big]}_{\text{ideal ensemble error}} \; + \; \underbrace{\mathbb{E}_X\left[\frac{\lambda\, D}{K}\, \delta(X)\right]}_{\text{OSR mismatch error}}$$

*where $\ell(\cdot)$ is $\lambda$-Lipschitz in its first argument, $D$ bounds expert disagreement, $\delta(X)$ denotes the selection mismatch size, and $\bar{f}(x)$ denotes the average predictor over the oracle-selected experts.*

**Interpretation.** The error of the ensemble predictor arises from two distinct sources. The first is the performance of the local experts themselves, captured by the oracle ensemble risk term, which reflects how well local training procedures approximate their respective optimal predictors. In practice, this term depends on the effectiveness of gradient-based optimization in nonconvex settings. The second source of error is induced by expert selection and is governed by the selection mismatch size $\delta(X)$. This term quantifies discrepancies between the experts selected by the learned mechanism and those selected by the oracle. Importantly, the bound shows that even under high client disagreement, the additional aggregation error vanishes if the selection mechanism consistently identifies the oracle experts. This highlights expert selection accuracy, rather than heterogeneity alone, as the primary bottleneck for ensemble-based aggregation, and motivates the design of improved Open Set Recognition (OSR) mechanisms to achieve robust performance in heterogeneous federated learning settings.

## 3.2 Insights from Extreme Label Skew Scenario

Under the worst-case error bounds derived above, the contrast between the two aggregation paradigms becomes clear. Parameter averaging exhibits an inherent sensitivity to client heterogeneity: when clients optimize different loss landscapes, the deviation from the centralized descent direction grows with both the degree of heterogeneity and the number of local update steps $T$. This sensitivity cannot be eliminated without restricting local optimization or increasing communication. In contrast, the ensemble-with-abstention approach can effectively manage extreme heterogeneity through accurate open-set recognition. When expert selection is sufficiently reliable, heterogeneity does not directly degrade performance. The sensitivity of the method is governed by the quality of the OSR mechanism, as captured by the leakage term, which ultimately determines the effectiveness of ensemble-based aggregation.

However, although the bound shows that heterogeneity can be mitigated through accurate OSR, greater heterogeneity can increase the intrinsic difficulty of the true OSR task itself (i.e requires more expressive or refined OSR mechanisms). To make this clearer, it is useful to look at an extreme but illustrative setting where the limitations of each approach become highly transparent. Consider the case where each client receives completely disjoint label support, so their information is entirely non overlapping. In such a regime, local optimization reveals the fundamental weaknesses of averaging and the role of OSR in restoring information. This issue is made precise in Proposition 1, whose proof is provided in the Appendix A.3.

**Proposition 1** (Mutual Information under Idealized Training and Label Skew). *Consider a classification task over $N$ labels with uniform class priors, $P(Y = y) = 1/N$. Each client is assigned a disjoint subset of $M$ labels, and models are trained under idealized conditions (perfect optimization, sufficient data). Let $Z_{\text{nosr}}$ denote the output of a model trained on disjoint labels without abstention, and let $Z_{\text{osr}}$ denote the output of a model trained with OSR, where clients abstain on out-of-distribution inputs. Then the mutual information between the model output and the true label satisfies*

$$\textit{Without OSR:} \qquad I(Z_{\text{nosr}}; Y) = \frac{M}{N} \log M,$$
$$\textit{With OSR:} \qquad I(Z_{\text{osr}}; Y) = \log N.$$

To interpret this result, let us look at the case where each client receives only a single class, $M = 1$. Then the information without OSR becomes $I(Z_{\text{nosr}}; Y) = 0$. This illustrates a fundamental limitation of aggregation

without abstention under extreme heterogeneity, including parameter averaging. If a client only ever sees examples from one label, it can minimize its loss by outputting that label constantly. Such a constant classifier contains no information about the input, and many such constant classifiers averaged together are still constant. Meanwhile, the global task still requires learning meaningful features for all classes. These two functions, a constant function and a feature-extracting function, are fundamentally incompatible. This is precisely what the drift term quantifies: local minimizers under extreme heterogeneity do not approximate the global minimizer in any meaningful sense.

On the other hand, the ensemble with abstention approach has no drift. By introducing a negative class or abstention signal, each client can explicitly mark points that fall outside its domain. This restores information that would otherwise be lost and allows the ensemble to represent all classes by concatenating the learned functions, not averaging them. However, when examined more closely, heterogeneity still matters by making accurate OSR more challenging. The mutual information result exposes the key mechanism: as heterogeneity increases and each client's label coverage shrinks, the amount of information missing from the local model, relative to the centralized optimum, increases. This gap must be recovered by the abstention mechanism. Formally, the centralized task achieves $I_{\text{cent}}(Y; Z) = \log N$, whereas a client with only $M$ labels can achieve at most $\frac{M}{N} \log M$. The information deficit created by heterogeneity is therefore $\Delta(M) = \log N - \frac{M}{N} \log M$. This quantity is monotonically decreasing in $M$: $\Delta(1) = \log N$ and $\Delta(N) = 0$. When $M = 1$, the OSR mechanism must recover the entire $\log N$ bits of information. When $M = N$, corresponding to the homogeneous case (assuming no proportion imbalance), there is no deficit to repair. Thus, even though the ensemble method does not suffer from optimization drift, the true open set recognition becomes harder as heterogeneity increases because the amount of missing information grows. The key distinction is that this effect is smaller since it does not destabilize optimization, and can be significantly reduced through better osr techniques and future innovations.

## 4    Methodology

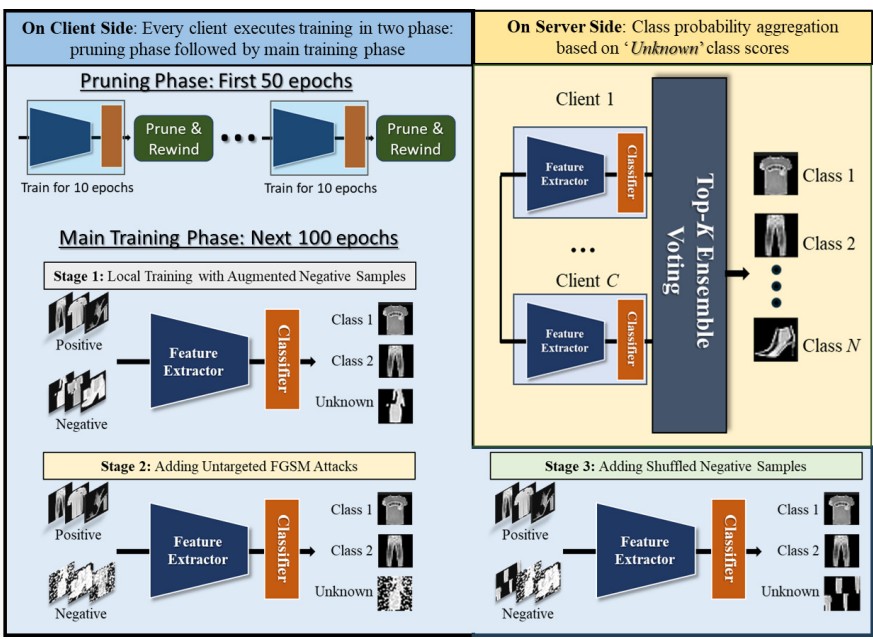

Figure 2: **Overview of FedSOV. Client-side:** Each client performs training in two phases, consisting of an initial *pruning phase* followed by a *main training phase*. During the main training phase, negative augmentation, adversarial perturbations, and shuffled negative samples are applied to enable open-set recognition. **Server-side:** The server aggregates client predictions using an ensemble voting mechanism based on unknown-class confidence scores to infer the final class label.

In this section, we introduce FedSOV and discuss the key motivations underlying the algorithm. These include an improved OSR mechanism designed to handle higher levels of heterogeneity, as well as a pruning strategy that addresses model size scalability challenge. Figure 1 provides a schematic overview of the method.

The pseudocode of our method is provided below to illustrate the main idea and overall workflow. For full implementation details and reproducibility, we refer the reader to the accompanying code repository.

---

**Algorithm 1** FEDSOV – Federated Scalable Open-Set Voting

---

    **Client-side execution (for all clients $i$ in parallel)**

1: **Client** $i$ **input:** local model $f_i(\cdot; \theta_i)$, initialization $\theta_i^0$, pruning epochs $E_{\text{prune}}$, main training epochs $E_{\text{main}}$, pruning stages $\mathcal{P}$, pruning ratio $p$, learning rate $\eta$

2: Initialize $\theta_i \leftarrow \theta_i^0$

    *Pruning phase: subnetwork selection via lottery ticket pruning*

3: Initialize pruning stage counter $k \leftarrow 1$

4: **for** $e = 1$ to $E_{\text{prune}}$ **do**

5:     Generate negative samples $\mathcal{D}_{i,\text{neg}} \leftarrow \text{NEGGEN}_{\text{all}}(\mathcal{D}_i^+)$

6:     $\theta_i \leftarrow \theta_i - \eta \nabla_\theta \mathcal{L}_{\text{osr}}(\theta_i; \mathcal{D}_i^+, \mathcal{D}_{i,\text{neg}})$

7:     **if** $e \bmod (E_{\text{prune}}/\mathcal{P}) = 0$ **and** $k \leq \mathcal{P}$ **then**

8:         $m_i^{(k)} \leftarrow \text{TopP}_{1-p}(|\theta_i|)$

9:         $m_i^{(k)} \leftarrow$ binary mask retaining the top $(1-p)$ fraction of weights by magnitude

10:         $\theta_i \leftarrow m_i^{(k)} \odot \theta_i^0$

11:         $k \leftarrow k + 1$

12:     **end if**

13: **end for**

    *Main training phase: open-set training of pruned models*

14: Initialize stage counter $s \leftarrow 1$

15: **for** $e = 1$ to $E_{\text{main}}$ **do**

16:     **if** $e \bmod (E_{\text{main}}/3) = 0$ **and** $s < 3$ **then**

17:         $s \leftarrow s + 1$

18:     **end if**

19:     **if** $s \geq 1$ **then**

20:         $\mathcal{D}_{i,\text{neg}}^{\text{aug}} \leftarrow \text{NEGGEN}_{\text{aug}}(\mathcal{D}_i^+)$

21:     **end if**

22:     **if** $s \geq 2$ **then**

23:         $\mathcal{D}_{i,\text{neg}}^{\text{fgsm}} \leftarrow \text{NEGGEN}_{\text{fgsm}}(\mathcal{D}_i^+)$

24:     **end if**

25:     **if** $s \geq 3$ **then**

26:         $\mathcal{D}_{i,\text{neg}}^{\text{shuffle}} \leftarrow \text{NEGGEN}_{\text{shuffle}}(\mathcal{D}_i^+)$

27:     **end if**

28:     $\mathcal{D}_{i,\text{neg}} \leftarrow \mathcal{D}_{i,\text{neg}}^{\text{aug}} \cup \mathcal{D}_{i,\text{neg}}^{\text{fgsm}} \cup \mathcal{D}_{i,\text{neg}}^{\text{shuffle}}$

29:     $\theta_i \leftarrow \theta_i - \eta \nabla_\theta \mathcal{L}_{\text{osr}}(\theta_i; \mathcal{D}_i^+, \mathcal{D}_{i,\text{neg}})$

30: **end for**

31: **Client output:** local model $f_i(\cdot; \theta_i)$

    **Server-side aggregation**

32: **Server input:** client models $\{f_i\}$

    *Top-K open-set voting based on confidence scores*

33: $S(x) \leftarrow \text{TopK}_i(\alpha_i(x))$

34: $f^*(x) \leftarrow \dfrac{1}{K} \sum_{i \in S(x)} f_i(x)$

35: **Server output:** global model $f^*$

---

**Operator definitions and notation.** $\mathcal{D}_i^+$: local labeled dataset available to client $i$. TopP: magnitude-based pruning retaining the largest-weight fraction. TopK: selection of the $K$ most confident clients. NegGen: local generation of synthetic negatives via augmentation, FGSM, and label shuffling. $\odot$: element-wise multiplication used to apply pruning masks to model parameters. $\alpha_i(x)$: confidence score produced by client model $f_i$ indicating the likelihood that input $x$ belongs to the client's known label space.

## 4.1 Designing the OSR Task

Open-set recognition (OSR) extends the standard classification setting by requiring a model to correctly classify samples from known classes while also rejecting inputs that do not belong to any known category. Let $\mathcal{K} = \{1, \ldots, K\}$ denote the set of known classes observed during training and let $\mathcal{U}$ denote the set of unknown classes that may appear at test time but are absent during training.

Formally, the OSR objective can be viewed as minimizing a combination of closed-set classification error and unknown detection error. Let $D_{\mathcal{K}}$ denote the distribution over known-class samples and $D_{\mathcal{U}}$ denote the distribution of unknown samples. The ideal OSR risk can then be written as

$$\mathcal{L}_{\text{OSR}} = \mathbb{E}_{(x,y) \sim D_{\mathcal{K}}} \left[ \ell_{\text{cls}}(f_\theta(x), y) \right] + \mathbb{E}_{x \sim D_{\mathcal{K}} \cup D_{\mathcal{U}}} \left[ \ell_{\text{det}}(f_\theta(x)) \right]$$

where the first term $\ell_{cls}$ corresponds to standard closed-set classification loss and the second term $\ell_{\text{det}}$ penalizes misclassification of unknown inputs as known classes or vice versa.

In practice, samples from $D_{\mathcal{U}}$ are not available during training. Consequently, practical OSR methods approximate this term through surrogate mechanisms such as confidence thresholding, extreme-value modeling, or synthetic negative sample generation. In the context of ensemble-based federated learning, reliable unknown detection is particularly important because abstention enables the ensemble to route inputs to the most appropriate client model. Following the approach introduced in FedOV, we approximate $D_{\mathcal{U}}$ using synthetic negative samples constructed directly in input space.

We implement OSR by introducing an explicit unknown class and training the model to map inputs that do not belong to any known class to this category. To this end, we construct synthetic negative samples that are intended to represent the unknown class. The objective is for the model to learn a decision boundary that separates known in-domain samples from inputs that lack the defining characteristics of those classes.

However, the effectiveness of this approach critically depends on how negative samples are constructed because discriminative models naturally focus on features that distinguish samples from one class from the others. Real-world images share a substantial set of class-agnostic features, such as spatial continuity, smoothness, local texture statistics, and low-level visual regularities. These shared features appear across all classes and should not be used as a signal for rejection. If negative samples violate these shared properties, a discriminative OSR classifier can easily rely on such artifacts as shortcut cues, rather than learning the absence of the semantic features that define a known class. Consequently, an effective OSR task must satisfy two principles: negative samples should preserve the shared structure present in real images, while selectively removing the class-specific features that characterize in-domain data. Under this construction, the only reliable distinction between positive and negative samples is the presence or absence of class-specific semantics, forcing the OSR classifier to focus on the causal features that define each class.

Existing constructions, such as the random cut-and-paste negatives used in FedOV, partially follow this intuition by disrupting global semantic structure. In this approach patches are extracted from an image and relocated or rotated. However, these operations also introduce shortcut cues in the form of abrupt patch boundaries, unnatural edges, and broken smoothness. Such artifacts do not correspond to any class-specific semantics, yet they violate the shared structure present in all real world iamges. As a result, the OSR classifier can easily distinguish positive and negative samples using noncausal cues, rather than learning the absence of class-specific features. To address this limitation, we introduce smoothed shuffled negatives, which eliminate global semantic structure while preserving shared visual statistics. Specifically, we partition the image into patches, apply a spatial permutation, and then smooth patch boundaries to remove permutation-induced discontinuities. This preserves local texture, color statistics, and continuity while destroying global class-specific structure, leading to improved performance in heterogenous settings.

In our method we include the negative samples of FedOV as well but avoid latent-space interpolation methods such as PROSER, as latent representations are often entangled and do not permit controlled removal of class-specific semantics. Such interpolations may also generate off-manifold samples that fail to reflect realistic unseen data, thereby encouraging shortcut discrimination. In contrast, our input-space construction provides direct and interpretable control over semantic destruction.

### 4.2 Pruning as a Solution to Scalability

While improved OSR design enhances predictive performance and robustness, a more fundamental challenge remains: scalability. As the number of clients increases, the number of local models grows accordingly, making inference increasingly costly. Prior work attempts to address this issue via knowledge distillation; however, distillation can incur information loss, exhibit instability under high data heterogeneity, and requires access to public server-side data, a constraint that is frequently unmet in federated settings. Motivated by these limitations, we explore pruning as an alternative and more direct approach to achieving scalability. Further discussion comparing these approaches can be found in Appendix B.4, C.2 and C.3.

Consider a global classification problem. Let $F^{task}$ denote the set of semantic features required to solve the problem. Suppose that a centralized model of size $\mathcal{S}$ is sufficient to represent $F^{task}$. The OSR ensemble provides an overcomplete representation of this feature set, since each client model contributes task-relevant structure and the ensemble achieves high accuracy. We therefore hypothesize that the ensemble admits a subnetwork of size $\mathcal{S}$ that preserves the essential semantic features needed for the task. Intuitively, if a model with capacity $\mathcal{S}$ can encode $F^{task}$, then pruning a successful ensemble down to size $\mathcal{S}$ should retain the feature components that carry predictive value. Pruning acts as a selection mechanism that extracts a compact representation of the ensemble's overcomplete feature basis.

This hypothesis is not guaranteed a priori, but we evaluate it empirically and find that, in settings where a single model of size $\mathcal{S}$ is not severely bottlenecked by dataset complexity, pruning the ensemble to size $\mathcal{S}$ matches and often exceeds the performance of the full ensemble. This behavior is consistent with the view that the ensemble contains redundant or overspecialized components and that a compact model that captures $F^{task}$ can generalize more effectively.

However, identifying a useful subnetwork within the ensemble requires a principled pruning strategy. Vanilla pruning techniques can lead to significant performance drops because naive compression or uniform magnitude pruning can remove rare but essential feature pathways and reduce discriminative capacity. We find that online pruning using a lottery-ticket style procedure Frankle & Carbin (2019) at the client level provides a more reliable mechanism for extracting subnetworks that retain, and often improve, test-time performance.

Moreover, when the centralized capacity $\mathcal{S}$ is insufficient for a given dataset, pruning offers a natural scalability dial: by reducing the pruning ratio, practitioners can allocate a larger model to capture more complex feature sets, trading off model size for accuracy in a controlled and interpretable manner. Thus, pruning is not only a mechanism for compressing OSR ensembles but also a capacity-tuning tool that adapts to dataset complexity and heterogeneity.

One important side note is that a pruned ensemble can theoretically achieve comparable or lower FLOPs and inference latency than a single distilled model, as discussed in Appendix C.2. However, achieving such gains in practice requires substantial systems-level optimization. In particular, without specialized sparse kernels and hardware-optimized inference implementations, zeroed weights are not skipped during computation, and thus do not translate into actual FLOP reduction, leaving wall-clock latency dominated by dense operations. In our experimental setup, we use standard PyTorch pruning, which zeros weights but does not reduce effective FLOPs during execution. Furthermore, our implementation does not exploit inter-branch parallelism, and therefore ensemble inference is slower than the distilled alternative in practice. As a result, the derivation in Appendix C.2 should be interpreted as a theoretical upper bound, illustrating that under ideal pruning and parallelism conditions, pruned ensembles can match or surpass monolithic models in inference speed. Realizing such gains in deployment, however, requires dedicated systems and hardware support.

# 5 Experiments

## 5.1 Experimental Setup

**Default Setup.** All models in the main experiments use a simple Convolutional Neural Network (CNN) with two convolutional layers and two fully connected layer. The learning rate was fixed at 0.001, following the configuration used in the FedOV implementation, and the batch size was set to 128. These values were chosen to balance training stability and computational efficiency given the large number of experimental settings. Regarding optimization, we followed the configurations commonly used for the respective algorithm families. Ensemble-based methods were trained using the Adam optimizer, following the setup used in the FedOV paper, while parameter-averaging methods were trained using SGD with momentum 0.9, consistent with the configuration used in the FedGF implementation. In preliminary pilot experiments we observed that parameter-averaging methods converged more reliably under SGD, whereas Adam provided stable training for ensemble-based models. All hyperparameters were kept identical across methods unless differences were required by the algorithm itself. Method-specific hyperparameters (e.g., those used in FedGF) were tuned through a small grid search. The complete list of hyperparameters and tuning ranges is provided in Appendix B.5. All experiments are run on a single NVIDIA RTX 4090 (24G) GPU.

**Baseline Methods:** We evaluate a diverse set of federated learning baselines spanning the two aggregation paradigms. For parameter averaging methods, we report FedAvg as the most commonly used baseline and FedGF as the strongest and most recent representative, based on our preliminary evaluations. For ensemble-based approaches, we compare FedSOV with FedOV in the main results, since FedOV represents the state of the art among ensemble methods under high heterogeneity. A broader set of classical, one-shot, and hybrid federated learning methods is evaluated in the Appendix B.1.

**Federated Configuration and Datasets:** Experiments are conducted on standard vision benchmarks. For parameter-averaging methods, we use 20 communication rounds with 5 local epochs per round on smaller datasets (MNIST, Fashion-MNIST, SVHN), and 100 communication rounds with 10 local epochs per round on larger datasets (CIFAR-10, CIFAR-100, Tiny-ImageNet). Ensemble-based methods are trained for a single communication round, using 10 local epochs on smaller datasets and 100 local epochs on larger datasets, resulting in substantially lower overall training compared to federated baselines. The main experiments evaluate 5, 10, and 20 client configurations under multiple heterogeneity regimes. These include highly non-IID Dirichlet partitions with concentration parameters $\alpha \in \{0.01, 0.1\}$ and an extreme label-disjoint setting. In the label-disjoint setting, clients observe non-overlapping class subsets. Let $C$ denote the number of clients and $N$ the number of classes. When $C \leq N$, the $N$ classes are evenly partitioned across clients such that each client receives $N/C$ exclusive classes. When $C > N$, each client is assigned samples from a single class, with samples evenly divided among clients corresponding to that class.This setting induces maximal distribution heterogeneity and serves as an extreme benchmark for federated learning methods. All results are averaged over three random seeds, with mean performance and standard deviation reported.

**Additional Experiments:** Beyond the main results, we study client scalability by evaluating a subset of configurations with 50 and 100 clients. We further analyze the effect of pruning by varying pruning ratios and pruning strategies. To assess the compression efficacy of pruning and distillation, we compare both compressed variants against the full ensemble. We also study feature skew using Colored MNIST, training parameter-averaging methods for 100 communication rounds with 5 local epochs per round and ensemble-based methods for a single round with 10 local epochs, evaluated across 5, 10, and 20 clients. Finally, we examine the robustness of the proposed approach across different CNN architectures. Appendix B presents additional results, including comprehensive comparisons with an expanded set of federated learning methods, as well as ablations over varying Dirichlet heterogeneity levels and server-side distillation data sizes.

## 5.2 Main Results

The main results are summarized in Tables 2, 3, and 4. Table 1 presents the centralized baseline accuracy, serving as an oracle-level reference point. Across all datasets and experimental settings, FedSOV consistently achieves the strongest overall performance, demonstrating robustness to data heterogeneity and favorable

scalability as the number of clients increases. This advantage holds across all three settings but is most pronounced under extreme heterogeneity, where client objectives are highly misaligned.

Table 1: Centralized (Oracle) Performance across Datasets

| MNIST | FMNIST | CMNIST | SVHN | CIFAR-10 | CIFAR-100 | Tiny-ImageNet |
|---|---|---|---|---|---|---|
| $99.20 \pm 0.14$ | $91.59 \pm 0.11$ | $98.94 \pm 0.05$ | $89.90 \pm 0.27$ | $70.43 \pm 0.34$ | $32.41 \pm 0.20$ | $17.17 \pm 0.60$ |

Table 2: Performance Comparison of Federated Learning Methods (Extreme Heterogeneity)

| Clients | Dataset | FedAvg | FedGF | FedOV | FedSOV* |
|---|---|---|---|---|---|
| 5 | MNIST | $72.33 \pm 0.95$ | $84.20 \pm 0.67$ | $83.80 \pm 1.36$ | $\mathbf{86.50 \pm 4.35}$ |
| | FMNIST | $59.90 \pm 0.26$ | $68.67 \pm 0.26$ | $67.16 \pm 2.02$ | $\mathbf{73.59 \pm 1.01}$ |
| | SVHN | $29.64 \pm 1.55$ | $21.02 \pm 0.08$ | $62.72 \pm 2.86$ | $\mathbf{71.34 \pm 1.41}$ |
| | CIFAR-10 | $48.30 \pm 0.68$ | $48.57 \pm 0.10$ | $44.92 \pm 1.35$ | $\mathbf{51.25 \pm 0.95}$ |
| | CIFAR-100 | $24.49 \pm 0.47$ | $26.79 \pm 0.53$ | $25.98 \pm 0.44$ | $\mathbf{28.67 \pm 0.24}$ |
| | Tiny-ImageNet | $12.12 \pm 0.10$ | $14.15 \pm 0.44$ | $10.48 \pm 0.17$ | $\mathbf{14.91 \pm 0.13}$ |
| 10 | MNIST | $30.68 \pm 3.28$ | $34.95 \pm 3.97$ | $76.48 \pm 0.89$ | $\mathbf{83.53 \pm 3.45}$ |
| | FMNIST | $37.28 \pm 4.22$ | $44.79 \pm 4.08$ | $60.10 \pm 2.83$ | $\mathbf{74.33 \pm 1.12}$ |
| | SVHN | $19.59 \pm 0.01$ | $19.59 \pm 0.01$ | $51.13 \pm 0.81$ | $\mathbf{73.17 \pm 0.24}$ |
| | CIFAR-10 | $22.17 \pm 1.16$ | $25.19 \pm 1.19$ | $29.91 \pm 0.82$ | $\mathbf{39.42 \pm 0.94}$ |
| | CIFAR-100 | $18.93 \pm 0.60$ | $19.12 \pm 0.04$ | $22.57 \pm 0.33$ | $\mathbf{25.46 \pm 0.31}$ |
| | Tiny-ImageNet | $9.75 \pm 0.15$ | $9.73 \pm 0.18$ | $9.23 \pm 0.21$ | $\mathbf{11.78 \pm 0.21}$ |
| 20 | MNIST | $24.85 \pm 3.42$ | $29.20 \pm 4.62$ | $84.85 \pm 1.45$ | $\mathbf{90.54 \pm 0.49}$ |
| | FMNIST | $37.42 \pm 6.25$ | $42.74 \pm 5.14$ | $70.70 \pm 0.30$ | $\mathbf{78.60 \pm 0.89}$ |
| | SVHN | $19.47 \pm 0.14$ | $19.55 \pm 0.06$ | $59.56 \pm 3.71$ | $\mathbf{71.17 \pm 0.40}$ |
| | CIFAR-10 | $22.96 \pm 0.63$ | $25.34 \pm 0.44$ | $39.35 \pm 0.74$ | $\mathbf{47.83 \pm 0.69}$ |
| | CIFAR-100 | $12.71 \pm 0.38$ | $12.37 \pm 0.32$ | $20.32 \pm 0.17$ | $\mathbf{23.88 \pm 0.41}$ |
| | Tiny-ImageNet | $6.38 \pm 0.16$ | $5.79 \pm 0.36$ | $8.04 \pm 0.23$ | $\mathbf{9.69 \pm 0.14}$ |

Performance Comparison under Non-IID Settings

Table 3: Dirichlet $\alpha = 0.01$

| Clients | Dataset | FedAvg | FedGF | FedOV | FedSOV* |
|---|---|---|---|---|---|
| 5 | MNIST | $71.67 \pm 3.28$ | $81.83 \pm 0.60$ | $85.93 \pm 3.72$ | $\mathbf{87.81 \pm 3.68}$ |
| | FMNIST | $54.04 \pm 5.67$ | $60.15 \pm 3.21$ | $66.68 \pm 4.84$ | $\mathbf{66.82 \pm 8.50}$ |
| | SVHN | $25.92 \pm 7.00$ | $20.82 \pm 1.59$ | $64.73 \pm 1.03$ | $\mathbf{72.83 \pm 4.81}$ |
| | CIFAR-10 | $47.53 \pm 2.68$ | $\mathbf{49.03 \pm 3.46}$ | $41.93 \pm 1.97$ | $48.50 \pm 1.75$ |
| | CIFAR-100 | $26.59 \pm 0.70$ | $27.44 \pm 0.71$ | $27.15 \pm 0.45$ | $\mathbf{30.30 \pm 0.43}$ |
| | Tiny-ImageNet | $13.70 \pm 0.39$ | $\mathbf{15.38 \pm 0.20}$ | $11.46 \pm 0.18$ | $15.28 \pm 0.50$ |
| 10 | MNIST | $47.39 \pm 5.22$ | $60.32 \pm 5.56$ | $\mathbf{61.33 \pm 4.31}$ | $54.15 \pm 8.30$ |
| | FMNIST | $39.93 \pm 6.61$ | $45.08 \pm 9.71$ | $\mathbf{59.69 \pm 1.26}$ | $58.97 \pm 2.76$ |
| | SVHN | $18.13 \pm 1.03$ | $19.57 \pm 0.03$ | $55.42 \pm 1.67$ | $\mathbf{63.52 \pm 6.43}$ |
| | CIFAR-10 | $31.90 \pm 5.80$ | $35.83 \pm 4.27$ | $35.55 \pm 3.48$ | $\mathbf{43.20 \pm 1.02}$ |
| | CIFAR-100 | $19.88 \pm 1.07$ | $19.35 \pm 0.90$ | $24.56 \pm 0.32$ | $\mathbf{27.47 \pm 0.40}$ |
| | Tiny-ImageNet | $10.75 \pm 0.23$ | $10.60 \pm 0.56$ | $10.46 \pm 0.10$ | $\mathbf{13.41 \pm 0.23}$ |
| 20 | MNIST | $31.00 \pm 2.15$ | $25.80 \pm 3.26$ | $74.95 \pm 6.61$ | $\mathbf{76.01 \pm 5.52}$ |
| | FMNIST | $36.04 \pm 3.76$ | $42.09 \pm 5.93$ | $\mathbf{62.83 \pm 1.60}$ | $59.52 \pm 2.94$ |
| | SVHN | $18.60 \pm 0.95$ | $19.58 \pm 0.01$ | $50.43 \pm 2.00$ | $\mathbf{52.66 \pm 9.21}$ |
| | CIFAR-10 | $26.22 \pm 2.45$ | $28.61 \pm 1.33$ | $34.87 \pm 2.02$ | $\mathbf{41.23 \pm 3.29}$ |
| | CIFAR-100 | $13.34 \pm 0.43$ | $12.50 \pm 0.59$ | $22.67 \pm 0.61$ | $\mathbf{25.09 \pm 0.36}$ |
| | Tiny-ImageNet | $7.31 \pm 0.27$ | $6.46 \pm 0.13$ | $8.83 \pm 0.25$ | $\mathbf{11.19 \pm 0.23}$ |

Table 4: Dirichlet $\alpha = 0.1$

| Clients | Dataset | FedAvg | FedGF | FedOV | FedSOV* |
|---|---|---|---|---|---|
| 5 | MNIST | $82.24 \pm 4.01$ | $86.32 \pm 5.08$ | $\mathbf{87.47 \pm 1.62}$ | $85.03 \pm 1.19$ |
| | FMNIST | $61.88 \pm 1.13$ | $67.44 \pm 0.65$ | $69.39 \pm 1.03$ | $\mathbf{69.70 \pm 2.14}$ |
| | SVHN | $32.45 \pm 8.10$ | $21.75 \pm 3.05$ | $72.19 \pm 3.99$ | $\mathbf{74.62 \pm 3.23}$ |
| | CIFAR-10 | $49.86 \pm 2.60$ | $52.55 \pm 0.37$ | $52.21 \pm 0.76$ | $\mathbf{52.78 \pm 1.49}$ |
| | CIFAR-100 | $28.51 \pm 0.76$ | $30.00 \pm 0.91$ | $29.01 \pm 0.14$ | $\mathbf{32.42 \pm 0.44}$ |
| | Tiny-ImageNet | $15.26 \pm 0.72$ | $\mathbf{16.85 \pm 0.59}$ | $11.63 \pm 0.14$ | $16.34 \pm 0.45$ |
| 10 | MNIST | $79.37 \pm 4.95$ | $84.50 \pm 2.80$ | $\mathbf{92.57 \pm 4.88}$ | $89.50 \pm 5.49$ |
| | FMNIST | $62.73 \pm 2.38$ | $67.13 \pm 2.56$ | $\mathbf{76.61 \pm 2.73}$ | $75.86 \pm 3.73$ |
| | SVHN | $20.23 \pm 0.50$ | $21.39 \pm 2.56$ | $75.54 \pm 2.94$ | $\mathbf{77.25 \pm 2.57}$ |
| | CIFAR-10 | $44.05 \pm 4.11$ | $43.59 \pm 4.46$ | $49.41 \pm 1.11$ | $\mathbf{50.66 \pm 0.84}$ |
| | CIFAR-100 | $23.97 \pm 0.64$ | $23.15 \pm 0.62$ | $27.21 \pm 0.43$ | $\mathbf{30.17 \pm 0.11}$ |
| | Tiny-ImageNet | $12.91 \pm 0.32$ | $12.60 \pm 0.37$ | $10.81 \pm 0.09$ | $\mathbf{13.92 \pm 0.40}$ |
| 20 | MNIST | $68.89 \pm 2.86$ | $66.88 \pm 2.78$ | $\mathbf{95.46 \pm 0.80}$ | $94.66 \pm 0.45$ |
| | FMNIST | $64.75 \pm 2.68$ | $63.02 \pm 3.14$ | $\mathbf{79.81 \pm 1.21}$ | $78.40 \pm 0.82$ |
| | SVHN | $19.57 \pm 0.06$ | $19.61 \pm 0.04$ | $70.64 \pm 0.69$ | $\mathbf{72.63 \pm 2.88}$ |
| | CIFAR-10 | $37.74 \pm 3.84$ | $42.40 \pm 2.10$ | $51.26 \pm 0.65$ | $\mathbf{54.75 \pm 0.64}$ |
| | CIFAR-100 | $17.87 \pm 1.00$ | $16.42 \pm 0.80$ | $24.22 \pm 0.48$ | $\mathbf{27.10 \pm 0.59}$ |
| | Tiny-ImageNet | $9.47 \pm 0.24$ | $8.26 \pm 0.15$ | $9.30 \pm 0.21$ | $\mathbf{11.77 \pm 0.15}$ |

From the above results we can see that as the degree of heterogeneity decreases, the performance gap narrows, and the ensemble baseline FedOV occasionally achieves comparable or slightly better results on simpler datasets. This is consistent with our theoretical analysis, which suggests that the benefits of improved open-set recognition become more critical as heterogeneity increases, while weaker OSR mechanisms may suffice in lower heterogeneity regimes. Similarly, FedGF can occasionally outperform FedSOV in low-heterogeneity, small-client settings, although these differences are generally minor. We further provide a detailed statistical significance analysis in the appendix to complement the main results. We observe that most improvements achieved by FedSOV are statistically significant, particularly in more challenging and heterogeneous regimes.

In contrast, we observe higher variance and occasional lack of statistical significance in low-client, low-heterogeneity settings on simpler datasets such as MNIST and FMNIST. This is expected due to performance saturation in these regimes, where differences between methods are inherently small, combined with the limited number of runs (three seeds). Establishing statistical significance in such settings would require a substantially larger number of trials. Overall, FedSOV demonstrates its strongest advantages under high heterogeneity, where client objectives are most misaligned. We also note the underperformance of parameter-averaging methods on SVHN in our setting; Appendix B.7 analyzes this failure mode in detail.

An important point to note here is model size. Although Fe-dOV demonstrates strong performance in some cases, its primary practical limitation is the large model footprint incurred by maintaining multiple experts. In contrast, FedSOV shows that this limitation can be effectively addressed through principled pruning. As shown in Table 5, the resulting FedSOV model has the same parameter count as parameter-averaging methods, while still achieving performance that exceeds that of FedOV.

Table 5: Parameter Count Comparison Across Methods

| Clients | FedAvg | FedGF | FedOV | FedSOV* |
|---------|--------|-------|-------|---------|
| 5       | 150K   | 150K  | 750K  | 150K    |
| 10      | 150K   | 150K  | 1.5M  | 150K    |
| 20      | 150K   | 150K  | 3M    | 150K    |

When comparing the two paradigms, ensemble-based methods exhibit substantially greater robustness to data heterogeneity by allowing client models to specialize, rather than enforcing a single global parameter consensus. This advantage is particularly pronounced in disjoint label-support scenarios, where parameter-averaging methods struggle to learn meaningful global representations. Beyond robustness, ensemble-based approaches are also more communication-efficient, as they require fewer training epochs and only a single communication round for model upload. This makes them especially well suited for communication-constrained federated learning settings. In contrast, parameter-averaging methods such as FedAvg and FedGF are highly sensitive to both heterogeneity severity and client count. As either increases, their performance degrades and they require substantially more training due to repeated communication rounds.

To make this comparison explicit, Table 6 reports the approximate communication volume across methods, computed as transmitted parameters (model size × communication rounds). Because FedSOV operates in a one-shot setting and maintains the same model size as parameter-averaging methods, it achieves substantially lower communication cost than both Fe-dAvg/FedGF and the ensemble-based FedOV approach.

| Method  | Rounds | Total Communication |
|---------|--------|---------------------|
| FedAvg  | 100    | 15M                 |
| FedGF   | 100    | 15M                 |
| FedOV   | 1      | 750K–3M             |
| FedSOV* | 1      | 150K                |

Table 6: Approximate communication volume across methods for CIFAR-10.

Next, we study the effect of the pruning ratio and different pruning strategies, as illustrated in Figure 3. When pruning is performed appropriately, increasing the pruning ratio does not degrade performance and can even lead to accuracy improvements. This behavior is consistent with recent work on ensemble pruning, which shows that while naive pruning strategies may fail, carefully designed pruning methods can improve out-of-distribution generalization Qiao & Peng (2024). As shown in the figure, when pruning is applied naively, performance degrades sharply beyond a certain pruning threshold, highlighting the importance of informed pruning strategies.

To evaluate the scalability of our method with respect to the number of clients, we consider high-client regimes with 50 and 100 clients on CIFAR-10 and CIFAR-100, with 10 and 100 total classes respectively. As the number of clients increases, the main results show that parameter-averaging methods degrade sharply, whereas ensemble-based approaches remain substantially more robust. This trend persists at very high client counts, as shown in Table 7, where parameter-averaging methods largely break down while ensemble methods continue to perform reliably. According to our theoretical analysis, this robustness can be attributed to stable open-set behavior, since the resulting error depends critically on the model's ability to

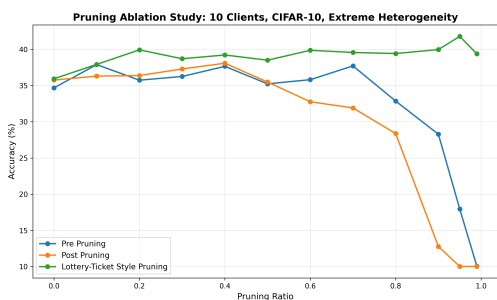

Figure 3: Pruning ratio up to 99% pruning.

handle open-set recognition. To further investigate this effect, we report open-set performance results in Appendix B.3 and analyze them in detail, providing additional insight into why the proposed method maintains strong performance at large client scales.

Table 7: High Client Count Performance Comparison

| Dataset | Heterogeneity | Clients | FedGF | FedOV | FedSOV* |
|---|---|---|---|---|---|
| **CIFAR-10** | Extreme Heterogeneity | 50 | 25.13 | 42.25 | **50.53** |
| | | 100 | 24.97 | 42.88 | **48.32** |
| | Non-IID (Dirichlet 0.1) | 50 | 38.29 | 49.16 | **53.04** |
| | | 100 | 31.92 | 45.56 | **51.20** |
| **CIFAR-100** | Extreme Heterogeneity | 50 | 6.16 | 15.66 | **19.89** |
| | | 100 | 1.38 | 12.01 | **18.14** |
| | Non-IID (Dirichlet 0.1) | 50 | 2.78 | 21.39 | **24.84** |
| | | 100 | 1.58 | 18.02 | **22.09** |

Table 8 shows the results in a feature-skew setting, where the underlying task is relatively simple (homogeneously split MNIST with only a color shift), and observe that parameter-averaging methods require substantially more training and many additional communication rounds to reach acceptable performance.

Due to the induced domain shift, low-round training is insufficient for these methods, often resulting in very low accuracy. Achieving reasonable performance therefore requires significantly more communication, up to 100 rounds, and even under this extended training regime performance degrades as the number of clients increases. In contrast, FedSOV achieves strong performance with substantially less training, a single communication round, and the same number of model parameters, highlighting its robustness to feature skew as well.

Table 8: Feature Skew Results on CMNIST

| Clients | FedAvg | FedGF | FedOV | FedSOV* |
|---|---|---|---|---|
| 5 | $95.91 \pm 0.84$ | $96.84 \pm 0.28$ | $98.38 \pm 0.14$ | $98.38 \pm 0.01$ |
| 10 | $95.11 \pm 0.38$ | $94.43 \pm 0.33$ | $97.78 \pm 0.13$ | $97.56 \pm 0.03$ |
| 20 | $75.43 \pm 7.23$ | $44.22 \pm 6.42$ | $97.31 \pm 0.22$ | $96.54 \pm 0.03$ |

Table 9: Comparison of Unpruned Ensembles, Distillation, and FedSOV Across Client Scales

| Dataset | Heterogeneity | Clients | Full Ensemble | Distilled Ensemble | FedSOV* |
|---|---|---|---|---|---|
| **CIFAR-10** | Extreme Heterogeneity | 5 | **52.20** | 46.30 | 51.25 |
| | | 10 | 37.94 | 28.69 | **39.42** |
| | | 20 | 47.20 | 43.81 | **47.83** |
| | | 50 | **51.66** | 49.94 | 50.53 |
| | | 100 | **52.07** | 49.51 | 48.32 |
| | Non-IID (Dirichlet 0.1) | 5 | **55.67** | 52.35 | 52.78 |
| | | 10 | 49.50 | 46.69 | **50.66** |
| | | 20 | **55.53** | 53.61 | 54.75 |
| | | 50 | 49.43 | 49.35 | **53.04** |
| | | 100 | 50.04 | 47.45 | **51.20** |
| **CIFAR-100** | Extreme Heterogeneity | 5 | 27.34 | 25.48 | **28.67** |
| | | 10 | 25.17 | 23.20 | **25.46** |
| | | 20 | **25.00** | 21.50 | 23.88 |
| | | 50 | **21.97** | 16.82 | 19.89 |
| | | 100 | **19.76** | 13.42 | 18.14 |
| | Non-IID (Dirichlet 0.1) | 5 | 29.46 | 28.37 | **32.42** |
| | | 10 | 29.34 | 28.28 | **30.17** |
| | | 20 | 25.99 | 25.51 | **27.10** |
| | | 50 | 22.93 | 22.48 | **24.84** |
| | | 100 | 21.62 | 20.83 | **22.09** |

Since the major bottleneck of any ensemble approach is the increasing model size, previous approaches have proposed distillation as the remedy. However the key limitation of distillation in federated learning is its reliance on server-side data, which is typically unavailable due to privacy constraints. Distillation is only feasible in semi-supervised federated learning settings where the server has access to unlabeled data and can train a student model using the soft predictions of the ensemble. To analyze whether better method of compression in this case is distillation or pruning, In Table 9, we directly compare the full ensemble with its

compressed variants obtained via distillation and pruning. To characterize an upper bound on distillation performance, we perform server-side distillation using IID data sampled from the same dataset. For all client configurations, the pruning ratio is adjusted to ensure an equal parameter budget across methods. The results show that pruning is not only more practical in federated learning, as it does not require server-side data, but also consistently yields stronger performance. In fact, the generalization benefits induced by pruning allow the resulting models to match or even exceed the performance of the full ensemble in many settings. One additional thing to note is that even in the semi supervised setting where data is available at server, the distilled student is constrained not only by the capacity of the ensemble but also by the information content of the available data, since only knowledge supported by the data distribution can be transferred. To study this limitation, we further investigate through additional experiments with varying data at the server to see performance in Appendix B.4.

The Simple CNN architecture is used for the majority of experiments because our evaluation requires a large experimental grid spanning multiple datasets, heterogeneity settings, and client counts. Simple CNN models train quickly while still achieving strong centralized performance on standard federated learning benchmarks such as CIFAR-10, CIFAR-100, SVHN, MNIST, and FMNIST, which are relatively small-scale datasets. This makes them well suited for extensive federated evaluations where many configurations must be tested. Nevertheless, to verify that the advantages of our method are not specific to the Simple CNN backbone, we conduct a targeted model ablation study. In this experiment, we evaluate LeNet, AlexNet, and a Large CNN on CIFAR-10 across multiple heterogeneity levels and client-count settings. For ensemble-based methods, local training is performed for 100 epochs, while the FedAvg baseline is trained for 20 rounds with 5 local epochs per round to maintain approximately comparable computational cost across methods. In Table 12 we note that absolute accuracy can sometimes decrease when using larger architectures in federated learning benchmarks. This behavior is expected because the amount of data available to each client is limited, particularly under heterogeneous partitions. Larger models have higher capacity and can therefore be more prone to overfitting on smaller per-client datasets, whereas simpler architectures such as Simple CNN often generalize more reliably in these settings.

Finally, we conduct additional ablations in Appendix B.2, including a learning-rate search for the FedAvg baseline, to account for the known sensitivity of parameter-averaging methods to optimizer hyperparameters.

Table 10: CIFAR10 – Extreme Heterogeneity

| Model | Clients | FedSOV | Distilled Ensemble | FedOV | FedAvg |
|---|---|---|---|---|---|
| Simple CNN | 10 | **39.42** | 28.69 | 29.91 | 18.53 |
| AlexNet | 10 | **33.61** | 27.37 | 25.02 | 11.79 |
| LeNet | 10 | **30.85** | 23.57 | 26.32 | 16.45 |
| Large CNN | 10 | **31.67** | 25.88 | 21.84 | 10.03 |
| Simple CNN | 20 | **47.34** | 43.81 | 39.31 | 18.22 |
| AlexNet | 20 | **41.33** | 29.86 | 31.78 | 10.75 |
| LeNet | 20 | 31.72 | 25.59 | **32.95** | 13.32 |
| Large CNN | 20 | **42.18** | 32.22 | 35.07 | 10.00 |

Table 11: CIFAR10 – Dirichlet 0.01

| Model | Clients | FedSOV | Distilled Ensemble | FedOV | FedAvg |
|---|---|---|---|---|---|
| Simple CNN | 10 | **45.52** | 32.23 | 38.54 | 26.00 |
| AlexNet | 10 | **35.52** | 30.04 | 26.10 | 12.53 |
| LeNet | 10 | **32.68** | 27.53 | 30.07 | 17.22 |
| Large CNN | 10 | **35.30** | 26.85 | 27.80 | 11.80 |
| Simple CNN | 20 | **37.08** | 33.17 | 32.02 | 27.00 |
| AlexNet | 20 | **37.59** | 30.14 | 30.91 | 15.83 |
| LeNet | 20 | **36.89** | 30.94 | 34.80 | 24.31 |
| Large CNN | 20 | **43.55** | 35.09 | 32.63 | 23.84 |

Table 12: CIFAR10 – Dirichlet 0.1

| Model | Clients | FedSOV | Distilled Ensemble | FedOV | FedAvg |
|---|---|---|---|---|---|
| Simple CNN | 10 | **52.87** | 46.69 | 50.29 | 46.37 |
| AlexNet | 10 | 50.09 | 49.82 | **50.22** | 29.42 |
| LeNet | 10 | **47.25** | 43.72 | 44.61 | 42.04 |
| Large CNN | 10 | **57.55** | 50.84 | 41.16 | 39.48 |
| Simple CNN | 20 | **54.74** | 53.61 | 51.20 | 42.04 |
| AlexNet | 20 | **53.48** | 44.33 | 50.28 | 24.60 |
| LeNet | 20 | **47.16** | 39.42 | 46.91 | 37.44 |
| Large CNN | 20 | **56.40** | 46.07 | 56.03 | 36.44 |

# 6    Conclusion

In this study we explored an alternative to the dominant parameter-averaging paradigm for federated learning under high heterogeneity, and showed both theoretically and empirically that ensemble aggregation with

abstention is better suited to addressing heterogeneity in a more communication-efficient manner. The main practical challenge is model size growth, which we address through FedSOV, showing that pruning can effectively control ensemble size. Our results suggest that improved aggregation and training strategies beyond parameter averaging can yield significant performance gains in highly heterogeneous settings. There is further potential to enhance performance by using improved OSR methods, more effective ensemble aggregation strategies, and more advanced pruning techniques.

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

## Appendix

**Contents**

## A  Proofs

### A.1  Theorem 1

We begin by stating the assumptions and restating the theorem, followed by the proof establishing the optimization error of parameter averaging with respect to the ideal descent step.

**Notation**  We denote by $L_c(\theta) = \mathbb{E}_{(x,y)\sim D_c}[\ell(f(x;\theta), y)]$ the expected loss of client $c$, where $\ell(\cdot, \cdot)$ denotes the per-sample loss and the expectation is taken over samples drawn from the true client data distribution $D_c$. $D$ denotes the global data distribution obtained as the uniform mixture of client distributions.

**Assumption** (Regularity Conditions). *The following conditions hold:*

(i) **Smoothness.** *Each local objective $L_c$ is $\beta$–smooth, i.e.,*

$$\|\nabla L_c(\theta) - \nabla L_c(\theta')\| \leq \beta \|\theta - \theta'\|.$$

(ii) **Bounded Gradient Dissimilarity.** *For all $\theta$ and all clients $c$,*

$$\|\nabla L_c(\theta) - \nabla L(\theta)\| \leq B_c.$$

(iii) **Lipschitz Global Loss.** *The global loss $L(\cdot)$ is $G$–Lipschitz, i.e.,*

$$|L(\theta) - L(\theta')| \leq G \|\theta - \theta'\|.$$

**Theorem** (One-Round Parameter Averaging Optimization Error). *Under the above assumptions, the global loss of the averaged model after $T$ local gradient descent steps satisfies, for $T > 0$,*

$$\mathcal{E}_{\text{avg}} \leq \underbrace{\mathbb{E}_{(x,y)\sim \mathcal{D}}\big[\ell\big(f_{\tilde{\theta}^{(T)}}(x), y\big)\big]}_{\text{centralized error}} + \underbrace{\frac{G\bar{B}}{\beta}\Big((1+\eta\beta)^T - 1\Big)}_{\text{parameter averaging drift}}. \tag{3}$$

*Proof.* The error of the averaged model $L(\bar{\theta}^{(T)})$ can be decomposed into the error of the ideal centralized solution and an additional drift term introduced by local optimization and parameter averaging. Specifically, by adding and subtracting the centralized loss, we write

$$L(\bar{\theta}^{(T)}) = \underbrace{L(\tilde{\theta}^{(T)})}_{\text{centralized error}} + \underbrace{\big(L(\bar{\theta}^{(T)}) - L(\tilde{\theta}^{(T)})\big)}_{\text{drift error}}. \tag{4}$$

To get the drift error we analyze the deviation induced by client heterogeneity by first bounding the gradient mismatch between local and centralized objectives. For any client $c$ at step $t$,

$$\nabla L_c(\theta_c^{(t)}) - \nabla L(\tilde{\theta}^{(t)}) = \nabla L_c(\theta_c^{(t)}) - \nabla L_c(\tilde{\theta}^{(t)}) + \nabla L_c(\tilde{\theta}^{(t)}) - \nabla L(\tilde{\theta}^{(t)}). \tag{5}$$

Applying the triangle inequality yields

$$\|\nabla L_c(\theta_c^{(t)}) - \nabla L(\tilde{\theta}^{(t)})\| \leq \|\nabla L_c(\theta_c^{(t)}) - \nabla L_c(\tilde{\theta}^{(t)})\| + \|\nabla L_c(\tilde{\theta}^{(t)}) - \nabla L(\tilde{\theta}^{(t)})\|. \tag{6}$$

By $\beta$-smoothness of $L_c$ and bounded gradient dissimilarity, we have

$$\|\nabla L_c(\theta_c^{(t)}) - \nabla L(\tilde{\theta}^{(t)})\| \leq \beta \|\theta_c^{(t)} - \tilde{\theta}^{(t)}\| + B_c. \tag{7}$$

Each client performs $T$ local gradient descent steps

$$\theta_c^{(t+1)} = \theta_c^{(t)} - \eta \nabla L_c(\theta_c^{(t)}), \qquad \theta_c^{(0)} = \theta^0,$$

while centralized gradient descent follows

$$\tilde{\theta}^{(t+1)} = \tilde{\theta}^{(t)} - \eta \nabla L(\tilde{\theta}^{(t)}), \qquad \tilde{\theta}^{(0)} = \theta^0.$$

Subtracting the two updates gives, for each client $c$,

$$\theta_c^{(t+1)} - \tilde{\theta}^{(t+1)} = (\theta_c^{(t)} - \tilde{\theta}^{(t)}) - \eta\big(\nabla L_c(\theta_c^{(t)}) - \nabla L(\tilde{\theta}^{(t)})\big). \tag{8}$$

For each client $c$, taking norms in equation 8 and applying the triangle inequality yields

$$\|\theta_c^{(t+1)} - \tilde{\theta}^{(t+1)}\| \leq \|\theta_c^{(t)} - \tilde{\theta}^{(t)}\| + \eta\|\nabla L_c(\theta_c^{(t)}) - \nabla L(\tilde{\theta}^{(t)})\|. \tag{9}$$

Using equation 7, we first obtain

$$\begin{aligned}
\|\theta_c^{(t+1)} - \tilde{\theta}^{(t+1)}\| &\leq \|\theta_c^{(t)} - \tilde{\theta}^{(t)}\| + \eta\big(\beta\|\theta_c^{(t)} - \tilde{\theta}^{(t)}\| + B_c\big) \\
&= (1 + \eta\beta)\|\theta_c^{(t)} - \tilde{\theta}^{(t)}\| + \eta B_c.
\end{aligned} \tag{10}$$

Define the average client deviation at step $t$ as

$$\Delta_t := \frac{1}{C} \sum_{c=1}^{C} \|\theta_c^{(t)} - \tilde{\theta}^{(t)}\|.$$

Averaging equation 10 over all clients yields

$$\begin{aligned}
\Delta_{t+1} &= \frac{1}{C} \sum_{c=1}^{C} \|\theta_c^{(t+1)} - \tilde{\theta}^{(t+1)}\| \\
&\leq \frac{1}{C} \sum_{c=1}^{C} \Big((1 + \eta\beta)\|\theta_c^{(t)} - \tilde{\theta}^{(t)}\| + \eta B_c\Big) \\
&= (1 + \eta\beta)\Delta_t + \eta\bar{B},
\end{aligned} \tag{11}$$

where

$$\bar{B} := \frac{1}{C} \sum_{c=1}^{C} B_c.$$

Since all models are initialized identically, we have

$$\theta_c^{(0)} = \tilde{\theta}^{(0)} = \theta^0 \quad \forall c,$$

which implies

$$\Delta_0 = 0.$$

Unrolling the recursion for $T > 0$ steps gives

$$\Delta_T \leq \eta\bar{B} \sum_{k=0}^{T-1} (1 + \eta\beta)^k. \tag{12}$$

Using the geometric series identity, we obtain

$$\Delta_T \leq \frac{\bar{B}}{\beta}\big((1 + \eta\beta)^T - 1\big). \tag{13}$$

—-

Next, define the deviation of the averaged iterate as

$$d_t := \|\bar{\theta}^{(t)} - \tilde{\theta}^{(t)}\|, \quad \text{where } \bar{\theta}^{(t)} := \frac{1}{C}\sum_{c=1}^{C}\theta_c^{(t)}.$$

Then,

$$
\begin{aligned}
d_t &= \left\|\bar{\theta}^{(t)} - \tilde{\theta}^{(t)}\right\| \\
&= \left\|\frac{1}{C}\sum_{c=1}^{C}\left(\theta_c^{(t)} - \tilde{\theta}^{(t)}\right)\right\| \\
&\leq \frac{1}{C}\sum_{c=1}^{C}\|\theta_c^{(t)} - \tilde{\theta}^{(t)}\| \\
&= \Delta_t,
\end{aligned}
\tag{14}
$$

where the inequality follows from the triangle inequality.

Therefore we obtain

$$d_T \leq \frac{\bar{B}}{\beta}\left((1+\eta\beta)^T - 1\right).\tag{15}$$

From the Lipschitz continuity of the global loss, for $T > 0$ we have

$$|L(\bar{\theta}^{(T)}) - L(\tilde{\theta}^{(T)})| \leq \frac{G\bar{B}}{\beta}\left((1+\eta\beta)^T - 1\right).\tag{16}$$

Therefore, the error of the averaged model $L(\bar{\theta}^{(T)})$ in equation 4 can be upper bounded as

$$\mathcal{E}_{\text{avg}} \leq \underbrace{L(\tilde{\theta}^{(T)})}_{\text{centralized error}} + \underbrace{\left|L(\bar{\theta}^{(T)}) - L(\tilde{\theta}^{(T)})\right|}_{\text{drift error}}.\tag{17}$$

Using the worst-case deviation from the centralized iterate derived above, the average error satisfies, for $T > 0$,

$$\boxed{\mathcal{E}_{\text{avg}} \leq \underbrace{\mathbb{E}_{(x,y)\sim\mathcal{D}}\left[\ell\left(f_{\tilde{\theta}^{(T)}}(x), y\right)\right]}_{\text{centralized error}} + \underbrace{\frac{G\bar{B}}{\beta}\left((1+\eta\beta)^T - 1\right)}_{\text{parameter averaging drift}}.}\tag{18}$$

$\square$

**Remark.** At the first local step, the averaged iterate coincides exactly with the centralized iterate:

$$\bar{\theta}^{(1)} = \tilde{\theta}^{(1)}.$$

This follows from the linearity of the averaging operation applied to the gradients:

$$\frac{1}{C}\sum_{c=1}^{C}\nabla L_c(\theta^0) = \nabla L(\theta^0).$$

As a result, the deviation at $t = 1$ satisfies

$$\|\bar{\theta}^{(1)} - \tilde{\theta}^{(1)}\| = 0.$$

This exact cancellation occurs only at the first step due to identical initialization. For $t \geq 1$, the local iterates diverge and such cancellation no longer holds. Consequently, the general bound derived above does not capture this initial alignment and is therefore conservative, particularly at small $T$.

## A.2 Theorem 2

In this section, we analyze the error of the ensemble predictor and characterize its dependence on open-set recognition (OSR) behavior. We first state the necessary definitions and assumptions, and then derive the error bound.

**Definition** (Abstention Scores). *Each client $c \in \mathcal{C} := \{1, \ldots, C\}$ outputs an abstention probability*

$$f_{\perp,c}(x) \in [0,1].$$

*The corresponding non-abstention (confidence) score is defined as*

$$a_c(x) := 1 - f_{\perp,c}(x) \in [0,1].$$

**Definition** (Oracle and Learned Expert Selection Sets). *Let $P(C = c \mid x)$ denote the oracle posterior relevance of expert $c$ for input $x$, and let $\tau > 0$ be a fixed relevance threshold. The oracle expert set is defined as*

$$S(x) := \{ c \in \mathcal{C} : P(C = c \mid x) \geq \tau \}.$$

*Let $a_c(x) \in [0,1]$ denote the non-abstention (confidence) score of expert $c$. The learned expert set is defined as*

$$\widehat{S}(x) := \left\{ c \in \mathcal{C} : a_c(x) \geq \text{Top-}K\big(\{a_{c'}(x)\}_{c' \in \mathcal{C}}\big) \right\},$$

*where $\text{Top-}K(\cdot)$ denotes the $K$-th largest value in the set.*

**Definition** (Correct, Incorrect, and Missed Expert Sets). *Define the correctly selected, incorrectly selected, and missed oracle expert sets as*

$$S_P(x) := S(x) \cap \widehat{S}(x), \qquad S_N(x) := \widehat{S}(x) \setminus S(x), \qquad S_M(x) := S(x) \setminus \widehat{S}(x).$$

*The* selection mismatch size *is defined as*

$$\delta(x) := |S_N(x)| = |S_M(x)|.$$

**Definition** (Matching of Mismatched Experts). *For any input $x \in \mathcal{X}$, since $|S_N(x)| = |S_M(x)|$, there exists a one-to-one matching*

$$\mathcal{T}(x) \subseteq S_N(x) \times S_M(x)$$

*such that each element of $S_N(x)$ and each element of $S_M(x)$ appears in exactly one pair $(n,p) \in \mathcal{T}(x)$. Consequently,*

$$|\mathcal{T}(x)| = \delta(x).$$

**Definition** (Learned Ensemble Predictors). *Let $f_c : \mathcal{X} \to \mathbb{R}^d$ denote the predictor associated with client $c \in \mathcal{C}$. The learned ensemble predictor is defined as*

$$f_{\text{ens}}(x) := \frac{1}{K} \sum_{c \in \widehat{S}(x)} f_c(x).$$

**Assumption** (Modeling Assumptions). *The following conditions hold:*

*(i) **Fixed Selection Cardinality.** For all inputs $x \in \mathcal{X}$, the oracle and learned expert sets satisfy*

$$|S(x)| = |\widehat{S}(x)| = K,$$

*for some fixed integer $K \geq 1$. This implies that for the oracle set, the threshold $\tau$ is non-static and satisfies $\tau(x) = P(C = \pi(K) \mid x)$, where $\pi(K)$ denotes the index of the $K$-th largest posterior.*

(ii) **Global Realizability.** *There exists a deterministic target function $y : \mathcal{X} \to \mathcal{Y}$ such that $Y = y(X)$ almost surely. Consequently, the population risk satisfies*

$$\mathbb{E}_{X,Y}[\ell(g(X), Y)] = \mathbb{E}_X[\ell(g(X), y(X))]$$

*for any predictor $g$.*

(iii) **Unbiased Noisy Bayes Estimators.** *There exists a Bayes-optimal predictor $f^* : \mathcal{X} \to \mathbb{R}^d$ such that for any $x \in \mathcal{X}$ and any oracle-selected expert $c \in S(x)$,*

$$f_c(x) = f^*(x) + \varepsilon_c(x),$$

*where the noise satisfies*

$$\mathbb{E}[\varepsilon_c(x)] = 0, \qquad \mathrm{Var}(\varepsilon_c(x)) = \sigma_c^2 \leq \sigma^2,$$

*and $\{\varepsilon_c(x)\}_{c \in S(x)}$ are conditionally independent given $x$.*

(iv) **Bounded Expert Disagreement.** *There exists a constant $D > 0$ such that for any $x \in \mathcal{X}$ and any experts $c, c' \in \mathcal{C}$,*

$$\|f_c(x) - f_{c'}(x)\| \leq D.$$

(v) **Lipschitz Loss.** *The loss function $\ell(\cdot)$ is $\lambda$-Lipschitz in its first argument, i.e., for all predictions $f(x), f'(x)$,*

$$\left|\ell(f(x)) - \ell(f'(x))\right| \leq \lambda \|f(x) - f'(x)\|.$$

**Theorem** (Optimization Error of the Learned Ensemble). *Under the definitions and assumptions stated above, the excess population risk of the learned ensemble predictor satisfies*

$$\mathcal{E}_{\mathrm{ens}} \leq \underbrace{\mathbb{E}_X\left[\ell(\bar{f}(X), y)\right]}_{\text{ideal ensemble error}} + \underbrace{\mathbb{E}_X\left[\frac{\lambda D}{K} \delta(X)\right]}_{\text{OSR mismatch error}}$$

*where $\ell(\cdot, y)$ is $\lambda$-Lipschitz in its first argument, $D$ bounds expert disagreement, $\delta(X)$ denotes the selection mismatch size, and $\bar{f}(x)$ denotes the average predictor over the oracle-selected experts.*

*Proof.* We begin by decomposing the population risk of the learned ensemble predictor $f_{\mathrm{ens}}$ as

$$\mathcal{L}(f_{\mathrm{ens}}) = \mathcal{L}(f_{\mathrm{ideal}}) + \left(\mathcal{L}(f_{\mathrm{ens}}) - \mathcal{L}(f_{\mathrm{ideal}})\right). \tag{19}$$

where $\mathcal{L}(f) := \mathbb{E}_X[\ell(f(X), y(X))]$ denotes the expected per-sample loss. Here, $f_{\mathrm{ideal}}$ denotes the oracle-selected ensemble predictor, assuming perfect knowledge of the relevance set $S(x)$ but no access to the Bayes predictor.

Taking absolute values and applying the triangle inequality yields

$$\mathcal{L}(f_{\mathrm{ens}}) \leq \mathcal{L}(f_{\mathrm{ideal}}) + |\mathcal{L}(f_{\mathrm{ens}}) - \mathcal{L}(f_{\mathrm{ideal}})|. \tag{20}$$

We next characterize the oracle risk term $\mathcal{L}(f_{\mathrm{ideal}})$. By Assumption (iii), for any input $x$ and any oracle-selected expert $c \in S(x)$, the local predictor can be written as the noisy estimate of the bayes optimal predictor. Therefore, the averaged oracle selected predictors is

$$\bar{f}(x) := \frac{1}{|S(x)|} \sum_{c \in S(x)} f_c(x). \tag{21}$$

Under the unbiasedness and independence in Assumption (iii), $\bar{f}(x)$ satisfies

$$\mathbb{E}[\bar{f}(x)] = f^*(x), \qquad \mathrm{Var}(\bar{f}(x)) \leq \frac{\sigma^2}{|S(x)|}.$$

Hence, $\bar{f}(x)$ is an unbiased, variance-reduced estimator of the Bayes-optimal predictor obtained by averaging oracle-selected experts. We therefore take $\bar{f}$ as the oracle reference predictor, noting that under Assumption (ii), where $Y$ is fully determined by the input $X$, and Assumption (iii) it is optimal among unbiased ensemble predictors formed by averaging oracle-selected experts. Consequently the population risk is given by

$$\mathcal{L}(f_{\text{ideal}}) = \mathbb{E}_X\left[\ell\big(\bar{f}(X), y\big)\right]. \tag{22}$$

We now bound the difference between the learned ensemble and the oracle ensemble at the prediction level. Using the disjoint decompositions we may write

$$\sum_{c \in \widehat{S}(x)} f_c(x) = \sum_{c \in S_P(x)} f_c(x) + \sum_{c \in S_N(x)} f_c(x), \tag{23}$$

and

$$\sum_{c \in S(x)} f_c(x) = \sum_{c \in S_P(x)} f_c(x) + \sum_{c \in S_M(x)} f_c(x). \tag{24}$$

Subtracting the two expressions yields

$$f_{\text{ens}}(x) - f_{\text{ideal}}(x) = \frac{1}{K}\left(\sum_{c \in S_N(x)} f_c(x) - \sum_{c \in S_M(x)} f_c(x)\right), \tag{25}$$

where the contributions from the correctly selected experts in $S_P(x)$ cancel. By definition Matching of Mismatched Experts, this difference can be rewritten as

$$f_{\text{ens}}(x) - f_{\text{ideal}}(x) = \frac{1}{K}\sum_{(n,p) \in \mathcal{T}(x)} \big(f_n(x) - f_p(x)\big). \tag{26}$$

Taking norms and applying the triangle inequality, we obtain

$$\big\|f_{\text{ens}}(x) - f_{\text{ideal}}(x)\big\| \leq \frac{1}{K}\sum_{(n,p) \in \mathcal{T}(x)} \big\|f_n(x) - f_p(x)\big\|. \tag{27}$$

By the Assumption (iv), $\|f_n(x) - f_p(x)\| \leq D$ for all $(n,p) \in \mathcal{T}(x)$, and hence

$$\big\|f_{\text{ens}}(x) - f_{\text{ideal}}(x)\big\| \leq \frac{D}{K}|\mathcal{T}(x)| = \frac{D}{K}\delta(x). \tag{28}$$

By Assumption (v), for any input $x \in \mathcal{X}$,

$$\big|\ell\big(f_{\text{ens}}(x)\big) - \ell\big(f_{\text{ideal}}(x)\big)\big| \leq \lambda\big\|f_{\text{ens}}(x) - f_{\text{ideal}}(x)\big\|. \tag{29}$$

Substituting the bound derived above yields

$$\big|\ell\big(f_{\text{ens}}(x)\big) - \ell\big(f_{\text{ideal}}(x)\big)\big| \leq \lambda\frac{D}{K}\delta(x). \tag{30}$$

Taking expectations over $X$ and combining with the earlier risk decomposition in 20, we obtain

$$\mathcal{L}(f_{\text{ens}}) \leq \mathcal{L}(f_{\text{ideal}}) + \mathbb{E}_X\left[\frac{\lambda D}{K}\delta(X)\right]. \tag{31}$$

By substituting 22 we obtain the final bound,

$$\boxed{\mathcal{E}_{\text{ens}} := \mathcal{L}(f_{\text{ens}}) \leq \mathbb{E}_X\left[\ell\big(\bar{f}(X), y\big)\right] + \mathbb{E}_X\left[\frac{\lambda D}{K}\delta(X)\right]} \tag{32}$$

$\square$

### A.3   Proposition 1

**Mutual Information without OSR**

We first specify the assumptions used to characterize the mutual information in the disjoint-label setting without open-set recognition.

**Notation.** $Y$ denotes the true label, and $Z$ denotes the label predicted by the client model.

**Definition** (Ensemble Output). *Let $Z_k$ denote the output random variable of client $k$. The ensemble output is defined as the joint random variable*

$$Z_{\text{ens}} := (Z_1, Z_2, \ldots, Z_K),$$

*where each $Z_k$ takes values in $\mathcal{Y}_k$ in the absence of open-set recognition, and in $\mathcal{Y}_k \cup \{\perp\}$ when open-set recognition (OSR) is enabled.*

**Assumption** (Idealized Disjoint-Label Setting without OSR). *The following conditions hold:*

*(i)* ***Uniform Label Prior.***
   *The global label space is $\mathcal{Y} = \{1, 2, \ldots, N\}$, and the true label $Y$ is drawn uniformly:*

$$P(Y = y) = \frac{1}{N}, \quad \forall y \in \mathcal{Y}.$$

*(ii)* ***Disjoint Label Partitions.***
   *Each client $k$ is assigned a subset $\mathcal{Y}_k \subset \mathcal{Y}$ such that*

$$|\mathcal{Y}_k| = M, \qquad \mathcal{Y}_i \cap \mathcal{Y}_j = \emptyset \text{ for } i \neq j, \qquad \bigcup_k \mathcal{Y}_k = \mathcal{Y}.$$

*(iii)* ***Perfect In-Distribution Prediction.***
   *For any label $y \in \mathcal{Y}_k$, the client model predicts deterministically:*

$$P(Z = y \mid Y = y) = 1.$$

*(iv)* ***Uniform Guessing on Out-of-Distribution Labels (No OSR).***
   *For $Y \notin \mathcal{Y}_k$, the model outputs a label from $\mathcal{Y}_k$ uniformly at random:*

$$P(Z = z \mid Y \notin \mathcal{Y}_k) = \frac{1}{M}, \quad \forall z \in \mathcal{Y}_k.$$

**Remark 1:** The disjoint-label setting considered above implicitly assumes that the number of clients does not exceed the number of classes. When the number of clients exceeds the number of labels, classes must be shared across multiple clients. In this case, the analysis does not apply verbatim at the level of individual clients; however, clients sharing the same label subset can be viewed collectively as a single effective unit. Under this interpretation, the core structure and intuition underlying the argument still holds.

**Remark 2:** Although Assumptions (iii) and (iv) may appear in tension, since perfect in-distribution prediction could induce systematic bias on out-of-distribution inputs, no such bias can be specified in the absence of explicit assumptions about inter-class similarity or feature overlap. Lacking principled knowledge of how out-of-distribution labels relate to in-support classes, we model this uncertainty using the principle of insufficient reason, leading to a uniform predictive distribution.

Given the above assumptions, the mutual information between the client output $Z$ and the true label $Y$ is

$$I(Z; Y) = \frac{M}{N} \log M.$$

**Proof**

By definition, the mutual information between $Z$ and $Y$ is

$$I(Z;Y) = H(Y) - H(Y \mid Z). \tag{33}$$

From Assumption (i), the label prior is uniform over $N$ classes, yielding

$$H(Y) = \log N. \tag{34}$$

Using the law of total conditional entropy, we expand

$$H(Y \mid Z) = \sum_{z \in \mathcal{Y}_k} P(Z = z) \, H(Y \mid Z = z) = -\sum_{z \in \mathcal{Y}_k} P(Z = z) \sum_{y \in \mathcal{Y}} P(Y = y \mid Z = z) \log P(Y = y \mid Z = z). \tag{35}$$

From Assumptions (iii) and (iv), the joint distribution $P(Y, Z)$ is given by

$$P(Y = y, Z = z) = \begin{cases} \frac{1}{N}, & y = z \in \mathcal{Y}_k, \\ \frac{1}{NM}, & y \notin \mathcal{Y}_k, \ z \in \mathcal{Y}_k, \\ 0, & \text{otherwise.} \end{cases} \tag{36}$$

The marginal distribution of $Z$ is obtained by marginalizing over $Y$:

$$\begin{aligned} P(Z = z) &= \sum_{y \in \mathcal{Y}} P(Y = y, Z = z) \\ &= \sum_{y \in \mathcal{Y}_k} P(Y = y, Z = z) + \sum_{y \notin \mathcal{Y}_k} P(Y = y, Z = z). \end{aligned} \tag{37}$$

Substituting from equation 36, for any $z \in \mathcal{Y}_k$ we obtain

$$P(Z = z) = \frac{1}{N} + (N - M) \cdot \frac{1}{NM} = \frac{1}{M}, \tag{38}$$

and hence $Z$ is uniformly distributed over $\mathcal{Y}_k$.

For a fixed prediction $z \in \mathcal{Y}_k$, the posterior distribution $P(Y \mid Z = z)$ follows from Bayes' rule:

$$P(Y = y \mid Z = z) = \frac{P(Y = y, Z = z)}{P(Z = z)}. \tag{39}$$

Using equation 36 and equation 38, we distinguish two cases:

(i) **In-distribution label $(y = z)$.**

$$P(Y = z \mid Z = z) = \frac{\frac{1}{N}}{\frac{1}{M}} = \frac{M}{N}. \tag{40}$$

(ii) **Out-of-distribution label $(y \notin \mathcal{Y}_k)$.**

$$P(Y = y \mid Z = z) = \frac{\frac{1}{NM}}{\frac{1}{M}} = \frac{1}{N}, \quad \forall y \notin \mathcal{Y}_k. \tag{41}$$

Using the posterior distribution derived above, the conditional entropy at a fixed prediction $Z = z \in \mathcal{Y}_k$ is

$$
\begin{aligned}
H(Y \mid Z = z) &= -\sum_{y \in \mathcal{Y}} P(Y = y \mid Z = z) \log P(Y = y \mid Z = z) \\
&= -P(Y = z \mid Z = z) \log P(Y = z \mid Z = z) - \sum_{y \notin \mathcal{Y}_k} P(Y = y \mid Z = z) \log P(Y = y \mid Z = z).
\end{aligned}
\tag{42}
$$

Substituting the posterior probabilities from equation 40 and equation 41, we obtain

$$
H(Y \mid Z = z) = -\frac{M}{N} \log\left(\frac{M}{N}\right) - (N - M) \cdot \frac{1}{N} \log\left(\frac{1}{N}\right).
\tag{43}
$$

Expanding the logarithms and simplifying yields

$$
\begin{aligned}
H(Y \mid Z = z) &= -\frac{M}{N}(\log M - \log N) + \left(1 - \frac{M}{N}\right) \log N \\
&= -\frac{M}{N} \log M + \frac{M}{N} \log N + \left(1 - \frac{M}{N}\right) \log N \\
&= \log N - \frac{M}{N} \log M.
\end{aligned}
\tag{44}
$$

Since all client outputs are symmetric under the assumptions and the ensemble output $Z_{\text{ens}} = (Z_1, \ldots, Z_K)$ does not reveal which client is informative in the absence of OSR, we apply the law of total conditional entropy:

$$
H(Y \mid Z_{\text{ens}}) = \sum_{k=1}^{K} P(K = k) \, H(Y \mid Z_k),
$$

where $K$ is a uniform random index over clients.

Because $P(K = k) = \frac{1}{K}$ and $H(Y \mid Z_k)$ is identical for all clients, this reduces to

$$
H(Y \mid Z_{\text{ens}}) = H(Y \mid Z_k).
$$

Since this value is the same for every $z \in \mathcal{Y}_k$ and $Z_k$ is uniformly distributed over $\mathcal{Y}_k$, we obtain

$$
H(Y \mid Z_{\text{ens}}) = \log N - \frac{M}{N} \log M.
$$

$\square$

### Mutual Information with OSR

We retain Assumptions (i)–(iii) from the disjoint-label setting and replace Assumption (iv) with the following:

**Assumption** (Deterministic Abstention on OOD Labels). *For any client $k$ and label $y \notin \mathcal{Y}_k$, the client abstains deterministically:*

$$
P(Z_k = \bot \mid Y = y) = 1, \qquad P(Z_k = z \mid Y = y) = 0, \quad \forall z \in \mathcal{Y}_k.
$$

**Remark.** Under this assumption, a client produces a non-abstaining output if and only if the true label lies in its assigned label subset. Combined with disjoint label partitions, this ensures that the ensemble output uniquely determines which client is informative for any given input.

**Proof**

We compute the conditional entropy of the true label given the ensemble output. By definition,

$$H(Y \mid Z_{\text{ens}}) = \sum_{z_{\text{ens}}} P(Z_{\text{ens}} = z_{\text{ens}}) \, H(Y \mid Z_{\text{ens}} = z_{\text{ens}}). \tag{45}$$

Fix any realizable ensemble output $Z_{\text{ens}} = z_{\text{ens}}$. By disjointness of the label partitions and deterministic abstention, there exists a unique label $y \in \mathcal{Y}$ consistent with $z_{\text{ens}}$. Consequently,

$$P(Y = y \mid Z_{\text{ens}} = z_{\text{ens}}) = 1, \tag{46}$$
$$P(Y = y' \mid Z_{\text{ens}} = z_{\text{ens}}) = 0, \quad \forall y' \neq y. \tag{47}$$

The conditional entropy for this realization is therefore

$$\begin{aligned} H(Y \mid Z_{\text{ens}} = z_{\text{ens}}) &= - \sum_{y' \in \mathcal{Y}} P(Y = y' \mid Z_{\text{ens}} = z_{\text{ens}}) \log P(Y = y' \mid Z_{\text{ens}} = z_{\text{ens}}) \\ &= -1 \cdot \log 1 \\ &= 0. \end{aligned} \tag{48}$$

Since equation 48 holds for all realizable ensemble outputs, substituting into equation 45 yields

$$H(Y \mid Z_{\text{ens}}) = 0. \tag{49}$$

Finally, the mutual information is

$$I(Z_{\text{ens}}; Y) = H(Y) - H(Y \mid Z_{\text{ens}}) = \log N. \tag{50}$$

$\square$

## B  Additional Experiments

### B.1  Extensive Federated Algorithm's Performance Comparison

Table 13: Performance comparison of all methods on CIFAR-10 and CIFAR-100 using Simple CNN, under extreme heterogeneity and Dirichlet $\alpha = 0.1$

| Method | CIFAR-10 | | CIFAR-100 | |
|---|---|---|---|---|
| | **Extreme** | $\alpha = 0.1$ | **Extreme** | $\alpha = 0.1$ |
| FedSOV | 39.42 | 50.66 | 25.46 | 30.17 |
| FedOV | 29.91 | 49.41 | 22.57 | 27.21 |
| FedGF | 25.19 | 43.59 | 19.12 | 23.15 |
| MoFedSAM | 26.26 | 44.92 | 18.66 | 22.16 |
| FedSAM | 26.56 | 45.90 | 18.39 | 23.21 |
| FedVLS | 8.08 | 51.44 | 18.88 | 29.16 |
| FedGH | 13.18 | 46.95 | 19.15 | 24.10 |
| FedAvg | 22.17 | 44.05 | 18.93 | 23.97 |
| SCAFFOLD | 20.90 | 42.63 | 19.65 | 25.24 |
| FedProx | 23.52 | 44.49 | 19.03 | 24.32 |
| FedMoon | 16.12 | 40.77 | 14.30 | 19.60 |
| FedDyn | 12.88 | 49.14 | 10.45 | 26.99 |
| FedConcat | 16.93 | 34.28 | 1.71 | 1.49 |
| FENS | 11.42 | 42.28 | 6.11 | 12.31 |
| CoBoosting | 10.84 | 21.9 | 3.15 | 4.50 |
| FuseFL(Resnet*) | 10.83 | 61.1 | 36.56 | 32.89 |

In Table 13, we observe that methods explicitly designed to emphasize consensus, such as gradient harmonization, perform poorly under extreme heterogeneity, despite being effective in more moderate heterogeneity settings. In this regime, there is little to no shared structure across clients for consensus-based mechanisms to exploit, and the problem manifests in its pure form: fully disjoint information where specialization is required rather than agreement. FedVLS similarly fails in this setting, as it relies on the global model to account for missing labels through distillation. When label supports are disjoint, the global model lacks any information about the missing classes, and the resulting distillation signal adversely affects local updates at each round, pushing models toward trivial optima. However, when some label overlap exists, FedVLS performs substantially better, as a limited degree of consensus can be established.

Classical optimization-based methods such as SCAFFOLD, FedProx, FedMoon, and FedDyn also struggle under extreme heterogeneity. In contrast, SAM-based methods demonstrate more robust performance, though ensemble-based approaches with explicit open-set recognition are clearly superior, particularly in the disjoint-label regime. FedSOV further resolves the scalability issue of ensemble methods through principled pruning, whereas FedOV scales linearly with the number of clients, resulting in a model that is C times larger than a single model. In our experiments, ensemble-based methods such as FedConcat, FENS, and Co-Boosting exhibit significant degradation under extreme heterogeneity. This suggests that, without mechanisms such as OSR, ensemble approaches may need to trade performance for communication efficiency in highly heterogeneous settings.

In our experiments, ensemble-based methods such as FedConcat, FENS, and Co-Boosting exhibit significant degradation under extreme heterogeneity. This suggests that, without mechanisms such as OSR, ensemble approaches may need to trade performance for communication efficiency in highly heterogeneous settings.

Co-Boosting relies on a static ensemble weighting scheme without input-dependent routing; although the weights are learned, they are shared across inputs, which limits its ability to adapt when different clients contain disjoint or client-specific information. FENS, on the other hand, relies on iterative parameter aggregation across multiple rounds and is therefore not strictly a one-shot method. While this design reduces computational cost by performing federated learning on a smaller network, its effectiveness depends on the quality of the learned representations. In extreme heterogeneity settings, as discussed in Proposition 1, if client backbones fail to learn sufficiently informative features, a lightweight aggregation network cannot recover meaningful global structure. For further comparison, we evaluate FENS alongside FedSOV and FedOV across different model architectures, as shown in Table 14. The table shows that FENS is less effective at handling heterogeneity compared to FedOV and FedSOV, particularly in highly heterogeneous settings. However, its performance approaches that of FedOV and FedSOV as the degree of homogeneity increases. This behavior can be attributed to the absence of OSR, which becomes increasingly important under stronger heterogeneity. Consistent with this, the gap between FedSOV and FedOV also widens in more heterogeneous regimes, further highlighting the importance of robust OSR mechanisms.

Table 14: Model-wise comparison of FedSOV, FedOV, and FENS on CIFAR-10 with 10 clients under Extreme Heterogeneity, Dirichlet $\alpha = 0.1$, and Homogeneous partitioning.

| Model | Extreme Heterogeneity | | | Dirichlet $\alpha = 0.1$ | | | Homogeneous | | |
|---|---|---|---|---|---|---|---|---|---|
| | FedSOV | FedOV | FENS | FedSOV | FedOV | FENS | FedSOV | FedOV | FENS |
| SimpleCNN | **39.42** | 29.91 | 11.42 | **52.87** | 50.29 | 42.28 | **65.13** | 64.66 | 62.20 |
| LeNet | **30.85** | 26.32 | 14.60 | **47.25** | 44.61 | 38.08 | **61.33** | 61.07 | 54.76 |
| AlexNet | **33.61** | 25.02 | 11.68 | 50.09 | **50.22** | 43.06 | 68.70 | **68.75** | 66.09 |
| LargeCNN | **31.67** | 21.84 | 11.76 | **57.55** | 41.16 | 46.98 | 73.07 | **74.42** | 68.79 |

We further note that the subpar performance of some of these methods may partly stem from sensitivity to hyperparameter choices. In particular, FedConcat and Co-Boosting involve several critical method-specific hyperparameters that require tuning across settings. For FedConcat, the number of client clusters must be specified a priori rather than learned adaptively. Since the optimal number of clusters is scenario-dependent, this rigidity likely contributes to its underperformance in our setting when using the default value of five clusters provided in the original implementation. Similarly, Co-Boosting involves multiple synthetic loss weights, generator configurations, and distillation parameters, which makes it sensitive to tuning and harder to adapt consistently across different regimes.

The FuseFL method is evaluated using a ResNet-18 backbone with 100 training epochs per layer-fusion stage, while the remaining baselines are trained using a Simple CNN. This is because the original FuseFL implementation relies on a progressive layer-wise fusion procedure in which intermediate feature blocks are concatenated and then frozen before training subsequent layers. Such layer-wise concatenation requires a customized architecture with clearly defined blocks and compatible feature dimensions. The Simple CNN architecture used by other methods does not support this mechanism without significant modification, so FuseFL is evaluated using its original ResNet-18 design. However, even with the advantages of a larger architecture and substantially more training epochs, FuseFL fails to perform well in settings where trivial solutions are readily available, such as the CIFAR-10 disjoint-class regime in which each client observes only a single class. In this scenario, learning a constant or degenerate predictor constitutes an easy local optimum, and the progressive fusion strategy is unable to recover meaningful cross-client structure. As a result, FuseFL is poorly suited for extreme heterogeneity settings where clients hold highly divergent and non-overlapping information.

## B.2 Extended Model Ablation Experiments

In the main results, we include a model ablation comparing FedSOV, distilled ensemble, FedOV, and FedAvg using the best-performing learning rate for FedAvg. In Table 15, we additionally report results for the full ensemble version of FedSOV using smoothed shuffled negatives, as well as multiple variants of FedAvg with different learning rates to further examine its sensitivity to this hyperparameter. Additionally, we extend

the model ablation to the FMNIST dataset in Table 18. Because FMNIST is a relatively easier dataset, we observe some cases where FedOV achieves the best performance, although our methods remain dominant in the majority of settings.

Table 15: Ablation results on CIFAR10 under Extreme Heterogeneity: test accuracy (%).

| Model | Clients | FedSOV | Full Ensemble | Distilled Ensemble | FedOV | FedAvg with lr 0.1 | FedAvg with lr 0.01 | FedAvg with lr 0.001 | FedAvg with lr 0.0001 |
|---|---|---|---|---|---|---|---|---|---|
| SimpleCNN2 | 10 | **39.42** | 37.94 | 28.69 | 29.91 | 26.01 | 18.53 | 14.84 | 14.92 |
| AlexNet | 10 | **33.61** | 31.88 | 27.37 | 25.02 | 11.73 | 11.79 | 11.56 | 10.00 |
| LeNet | 10 | **30.85** | 30.52 | 23.57 | 26.32 | 26.16 | 16.45 | 12.95 | 14.76 |
| LargeCNN | 10 | **31.67** | 28.13 | 25.88 | 21.84 | 10.28 | 10.03 | 11.49 | 10.03 |
| SimpleCNN2 | 20 | 47.34 | **47.79** | 43.81 | 39.31 | 27.74 | 18.22 | 15.55 | 10.60 |
| AlexNet | 20 | 41.33 | **41.84** | 29.86 | 31.78 | 10.32 | 10.75 | 10.00 | 10.00 |
| LeNet | 20 | 31.72 | **37.59** | 25.59 | 32.95 | 17.46 | 13.32 | 14.47 | 10.10 |
| LargeCNN | 20 | 42.18 | **42.29** | 32.22 | 35.07 | 12.40 | 10.00 | 11.05 | 10.00 |

Table 16: Ablation results on CIFAR10 under Dirichlet 0.01: test accuracy (%).

| Model | Clients | FedSOV | Full Ensemble | Distilled Ensemble | FedOV | FedAvg with lr 0.1 | FedAvg with lr 0.01 | FedAvg with lr 0.001 | FedAvg with lr 0.0001 |
|---|---|---|---|---|---|---|---|---|---|
| SimpleCNN2 | 10 | **45.52** | 41.27 | 32.23 | 38.54 | 28.59 | 26.00 | 16.13 | 14.34 |
| AlexNet | 10 | 35.52 | **38.17** | 30.04 | 26.10 | 13.08 | 12.53 | 11.15 | 10.00 |
| LeNet | 10 | 32.68 | **33.64** | 27.53 | 30.07 | 29.50 | 17.22 | 15.44 | 14.52 |
| LargeCNN | 10 | **35.30** | 34.02 | 26.85 | 27.80 | 19.96 | 11.80 | 12.14 | 10.30 |
| SimpleCNN2 | 20 | 37.08 | **46.32** | 33.17 | 32.02 | 32.38 | 27.00 | 19.90 | 12.71 |
| AlexNet | 20 | 37.59 | **37.92** | 30.14 | 30.91 | 20.55 | 15.83 | 10.00 | 10.00 |
| LeNet | 20 | **36.89** | 36.26 | 30.94 | 34.80 | 31.34 | 24.31 | 16.08 | 10.07 |
| LargeCNN | 20 | 43.55 | **44.05** | 35.09 | 32.63 | 24.68 | 23.84 | 17.14 | 11.26 |

Table 17: Ablation results on CIFAR10 under Dirichlet 0.1: test accuracy (%).

| Model | Clients | FedSOV | Full Ensemble | Distilled Ensemble | FedOV | FedAvg with lr 0.1 | FedAvg with lr 0.01 | FedAvg with lr 0.001 | FedAvg with lr 0.0001 |
|---|---|---|---|---|---|---|---|---|---|
| SimpleCNN2 | 10 | 52.87 | **53.79** | 46.69 | 50.29 | 10.00 | 46.37 | 25.40 | 18.23 |
| AlexNet | 10 | 50.09 | **51.80** | 49.82 | 50.22 | 43.48 | 29.42 | 11.15 | 10.00 |
| LeNet | 10 | 47.25 | 47.24 | 43.72 | 44.61 | **47.93** | 42.04 | 22.07 | 13.84 |
| LargeCNN | 10 | **57.55** | 55.37 | 50.84 | 41.16 | 57.41 | 39.48 | 22.27 | 10.05 |
| SimpleCNN2 | 20 | **54.74** | 53.75 | 53.61 | 51.20 | 10.00 | 42.04 | 24.59 | 10.01 |
| AlexNet | 20 | 53.48 | **54.60** | 44.33 | 50.28 | 10.00 | 24.60 | 10.00 | 10.00 |
| LeNet | 20 | 47.16 | **49.11** | 39.42 | 46.91 | 44.86 | 37.44 | 16.72 | 10.00 |
| LargeCNN | 20 | 56.40 | **57.47** | 46.07 | 56.03 | 10.00 | 36.44 | 16.95 | 8.47 |

Table 18: Ablation results on fashion-mnist under Extreme Heterogeneity: test accuracy (%).

| Model | Clients | FedSOV | Full Ensemble | Distilled Ensemble | FedOV |
|---|---|---|---|---|---|
| SimpleCNN2 | 10 | **74.87** | 71.03 | 39.27 | 63.11 |
| AlexNet | 10 | 66.07 | **66.93** | 50.66 | 66.93 |
| LeNet | 10 | 62.01 | 62.04 | 60.51 | **63.63** |
| LargeCNN | 10 | 53.35 | **57.56** | 34.55 | 53.38 |
| SimpleCNN2 | 20 | **78.36** | 77.58 | 34.12 | 71.17 |
| AlexNet | 20 | 65.36 | **68.18** | 40.29 | 68.08 |
| LeNet | 20 | 65.41 | **70.50** | 43.31 | 66.69 |
| LargeCNN | 20 | **66.63** | 59.61 | 31.23 | 60.51 |

Table 19: Ablation results on fashion-mnist under Dirichlet 0.01: test accuracy (%).

| Model | Clients | FedSOV | Full Ensemble | Distilled Ensemble | FedOV |
|-------|---------|--------|---------------|--------------------|-------|
| SimpleCNN2 | 10 | **66.72** | 65.58 | 64.49 | 66.48 |
| AlexNet | 10 | **57.20** | 55.08 | 50.38 | 54.14 |
| LeNet | 10 | 60.96 | 59.79 | 55.61 | **61.88** |
| LargeCNN | 10 | 59.83 | 60.06 | 52.13 | **64.74** |
| SimpleCNN2 | 20 | 65.25 | **66.32** | 64.78 | 62.86 |
| AlexNet | 20 | 59.65 | **60.17** | 54.94 | 54.98 |
| LeNet | 20 | 58.39 | **60.38** | 60.00 | 55.89 |
| LargeCNN | 20 | 57.15 | **63.80** | 54.35 | 60.76 |

Table 20: Ablation results on fashion-mnist under Dirichlet 0.1: test accuracy (%).

| Model | Clients | FedSOV | Full Ensemble | Distilled Ensemble | FedOV |
|-------|---------|--------|---------------|--------------------|-------|
| SimpleCNN2 | 10 | 74.60 | **77.51** | 76.46 | 76.44 |
| AlexNet | 10 | 69.62 | 70.49 | **73.90** | 70.78 |
| LeNet | 10 | 72.57 | **75.03** | 68.78 | 72.75 |
| LargeCNN | 10 | **80.55** | 76.23 | 70.78 | 78.45 |
| SimpleCNN2 | 20 | 80.69 | **80.79** | 77.61 | 79.74 |
| AlexNet | 20 | 71.06 | 71.42 | 70.78 | **74.09** |
| LeNet | 20 | **74.40** | 71.40 | 68.75 | 71.59 |
| LargeCNN | 20 | 76.84 | 72.34 | 75.06 | **77.69** |

## B.3 FedSOV Additional Ablation Experiments

To evaluate the robustness of our proposed method across different degrees of data heterogeneity, we conduct experiments by varying the Dirichlet concentration parameter from 0.01 to 0.9. This range spans highly heterogeneous to near-homogeneous data partitions, allowing us to assess performance sensitivity to heterogeneity as well as behavior in the homogeneous regime.

Table 21: Performance of FedSOV under different Dirichlet $\alpha$ values on CIFAR-10 and CIFAR-100

| Dataset | $\alpha = 0.01$ | $\alpha = 0.05$ | $\alpha = 0.1$ | $\alpha = 0.5$ | $\alpha = 0.7$ | $\alpha = 0.8$ | $\alpha = 0.9$ |
|---------|-----------------|-----------------|----------------|----------------|----------------|----------------|----------------|
| CIFAR-10 | 41.56 | 50.92 | 52.05 | 64.23 | 64.91 | 64.62 | 64.17 |
| CIFAR-100 | 27.74 | 29.17 | 30.48 | 30.57 | 30.52 | 29.65 | 30.50 |

We further ablate the effect of latent-space interpolation under extreme label skew by evaluating PROSER on the full ensemble. Using the unpruned ensemble isolates the impact of latent interpolation from pruning effects and allows us to assess whether our open-set recognition (OSR) benefit from latent-space interpolation. As shown in Table 22, PROSER does not yield consistent performance improvements in this regime, indicating that latent interpolation is ineffective when severe label imbalance is the dominant source of heterogeneity.

Table 22: Effect of PROSER on FedSOV under extreme heterogeneity

| Dataset | FedSOV(Unpruned) | FedSOV(Unpruned)-Proser |
|---------|------------------|-------------------------|
| CIFAR-10 | 37.94 | 35.96 |
| CIFAR-100 | 25.17 | 23.88 |

Finally, we ablate open-set recognition (OSR) performance to understand why our method remains effective in high-client regimes. Our theoretical analysis indicates that overall performance is closely tied to OSR quality, making its behavior across different client counts particularly informative. We report results for FedOV, FedSOV, and the full ensemble, which corresponds to the unpruned FedSOV varian.The comparison of FedOV and the full Ensemble result directly shows theadvantage of our smoothed shuffled negatives, as that is the main difference between these two methods.

A caveat in interpreting these results is that classification accuracy serves only as a proxy for OSR performance and does not directly measure how well the constructed task reflects true open-set conditions. This is especially relevant in the CIFAR-100 setting, where increasing the number of clients reduces the number of classes per client, gradually simplifying the task. In the extreme case of one class per client, OSR effectively becomes a binary decision, and higher accuracy does not necessarily imply stronger OSR capability. Despite this limitation, the results show that OSR performance remains robust across all client counts, and according to our theoretical analysis this explains the consistent scalability of our method. In contrast, parameter-averaging methods degrade sharply as the number of clients increases, even under relatively mild heterogeneity.

Table 23: Open-Set AUROC under Extreme Label Skew

| # Clients | Dataset | FedOV | Full Ensemble | FedSOV |
|---|---|---|---|---|
| 5 | CIFAR-10 | 0.6983 | 0.7425 | 0.7360 |
| | CIFAR-100 | 0.6149 | 0.6178 | 0.6211 |
| 10 | CIFAR-10 | 0.6985 | 0.7373 | 0.7551 |
| | CIFAR-100 | 0.6377 | 0.6486 | 0.6479 |
| 20 | CIFAR-10 | 0.7211 | 0.7652 | 0.7587 |
| | CIFAR-100 | 0.6724 | 0.6960 | 0.6927 |
| 50 | CIFAR-10 | 0.7246 | 0.7640 | 0.7663 |
| | CIFAR-100 | 0.7153 | 0.7572 | 0.7563 |
| 100 | CIFAR-10 | 0.7204 | 0.7554 | 0.7575 |
| | CIFAR-100 | 0.7527 | 0.7939 | 0.7974 |

## B.4 Distillation Ablations

A common approach for compressing ensembles is knowledge distillation, which transfers the ensemble's predictions to a single student model. However, distillation fundamentally requires data at the server, and the quality of the resulting student is inherently limited by the amount and diversity of this server-side data. If the public data is unrepresentative, the distilled model cannot capture knowledge from client distributions absent in the server data.

In the main experiments, we report an oracle distillation baseline that assumes access to the union of all clients' test data at the server. While this provides an optimistic upper bound, it obscures a critical limitation of distillation in federated settings. To isolate this effect, we ablate distillation performance with respect to both data diversity and data quantity available at the server. The results of these experiments show that as the diversity of server-side data decreases, the student model rapidly loses performance, indicating a failure to transfer the ensemble's knowledge. Similarly, reducing the total amount of available data leads to substantial degradation in accuracy. These observations confirm that the performance of the distilled student is capped not only by the quality of the ensemble but also, and often more severely, by the coverage of the server-side dataset. This dependence raises a fundamental question: if high-quality, representative data is available at the server, why not train a centralized model directly? Indeed, the only plausible scenario where distillation remains meaningful is when the server has access to unlabeled data drawn from the same distribution as the clients. Even in this favorable setting, however, distillation consistently underperforms pruning-based approaches in our experiments.

We further ablate the effect of the data source used for distillation and observe that distilling on the same data used by clients during training yields better performance than using different data from the same distribution. This is expected, as the ensemble produces more accurate and less noisy soft labels on data it has already seen. However, this setting constitutes a clear violation of the federated learning paradigm,

arguably more severe than assuming i.i.d. data at the server, since it presumes direct access to client training data.

Overall, these results highlight a key limitation of distillation in federated learning: its effectiveness is tightly coupled to server-side data availability, which is often unrealistic or incompatible with privacy constraints. In contrast, pruning offers a data-free compression mechanism, making it a more principled and practical alternative for scaling ensemble-based federated learning methods.

Table 24: Ablation: Number of Distillation Clients (Train Source)

| Heterogeneity | 10 Clients | 9 Clients | 8 Clients | 7 Clients | 6 Clients | 5 Clients | 4 Clients | 3 Clients | 2 Clients | 1 Client |
|---|---|---|---|---|---|---|---|---|---|---|
| Extreme Heterogeneity | 31.55 | 31.16 | 27.82 | 27.08 | 25.93 | 25.64 | 23.31 | 21.25 | 20.86 | 12.66 |
| Non-IID (Dirichlet 0.1) | 48.09 | 46.66 | 44.95 | 44.29 | 43.13 | 42.24 | 39.68 | 39.19 | 33.18 | 23.75 |

Table 25: Ablation: Number of Distillation Clients (Test Source)

| Heterogeneity | 10 Clients | 9 Clients | 8 Clients | 7 Clients | 6 Clients | 5 Clients | 4 Clients | 3 Clients | 2 Clients | 1 Client |
|---|---|---|---|---|---|---|---|---|---|---|
| Extreme Heterogeneity | 29.02 | 29.11 | 25.11 | 23.88 | 22.96 | 20.49 | 21.92 | 19.99 | 16.60 | 11.38 |
| Non-IID (Dirichlet 0.1) | 46.63 | 45.41 | 43.53 | 42.85 | 41.05 | 39.66 | 38.31 | 36.55 | 28.74 | 17.38 |

Table 26: Ablation: Distillation Data Ratio (Train Source)

| Heterogeneity | 1.0 | 0.75 | 0.5 | 0.25 | 0.1 |
|---|---|---|---|---|---|
| Extreme Heterogeneity | 31.55 | 31.63 | 28.25 | 28.98 | 27.74 |
| Non-IID (Dirichlet 0.1) | 48.09 | 45.94 | 45.33 | 43.03 | 41.01 |

Table 27: Ablation: Distillation Data Ratio (Test Source)

| Heterogeneity | 1.0 | 0.75 | 0.5 | 0.25 | 0.1 |
|---|---|---|---|---|---|
| Extreme Heterogeneity | 29.02 | 29.71 | 27.62 | 23.84 | 22.02 |
| Non-IID (Dirichlet 0.1) | 46.63 | 45.20 | 42.65 | 37.34 | 31.61 |

## B.5 FedGF Hyperparameter Ablation

FedGF introduces several method-specific hyperparameters that control the strength of global perturbations and the detection of client divergence during training. To ensure a fair comparison with our method, we performed a small hyperparameter search over the three primary FedGF parameters: the global perturbation strength $\rho$, the divergence threshold $T_D$, and the divergence tracking window size $W$.

Specifically, $\rho$ controls the magnitude of the global perturbation applied during the FedGF update, $T_D$ defines the threshold used to detect divergence between local and global updates, and $W$ specifies the number of rounds used to track divergence statistics. These parameters influence the stability of the training process as well as the degree of correction applied to mitigate client drift.

We conducted the ablation study on CIFAR-10 using the SimpleCNN2 architecture with 10 clients, 20 communication rounds, and 5 local epochs per round. The learning rate for the correction component was fixed to 0.001. Table 28 reports the resulting accuracies across three heterogeneity regimes (Dirichlet $\alpha = 0.1$, Dirichlet $\alpha = 0.01$, and the extreme label-skew setting).

Based on this search, we selected the configuration ($\rho = 0.05$, $T_D = 0.2$, $W = 30$) for the main experiments. This setting provided stable performance across heterogeneity regimes while avoiding the instability observed for larger perturbation strengths or extreme divergence thresholds. All FedGF results reported in the main paper use this configuration.

## B.6 Statistical significance of main results

In the main experiments we reported the mean test accuracy along with standard deviation across multiple random seeds. To further assess whether the observed performance differences are statistically significant, we conduct paired $t$-tests across seeds comparing FedSOV* against the baseline methods (FedOV, FedGF, and FedAvg).

For each dataset, client configuration, and heterogeneity setting, we compute the paired $t$-test using the accuracy values obtained from the same random seeds for both methods. Table 29 reports the resulting

Table 28: FedGF hyper-parameter ablation on CIFAR-10 (SimpleCNN2, clients=10, rounds=20, local epochs=5, cd_lr=0.001). Baseline: ($\rho$=0.05, $T_D$=0.2, $W$=30).

| $\rho$ | $T_D$ | $W$ | Dir 0.1 | Dir 0.01 | XtremeHetero |
|---|---|---|---|---|---|
| 0.05 | 0.2 | 30 | 25.10 | 18.04 | **17.68** |
| 0.0 | 0.0 | 0 | **25.44** | 16.10 | 14.85 |
| 0.0 | 0.0 | 30 | 11.66 | 11.78 | 13.35 |
| 0.0 | 0.0 | 100 | 11.70 | 14.06 | 13.13 |
| 0.0 | 0.05 | 0 | 25.41 | 16.14 | 14.96 |
| 0.0 | 0.05 | 30 | 11.78 | 12.08 | 13.05 |
| 0.0 | 0.05 | 100 | 11.58 | 11.71 | 14.65 |
| 0.0 | 0.5 | 0 | 25.38 | 16.01 | 15.04 |
| 0.0 | 0.5 | 30 | 25.40 | 16.13 | 14.84 |
| 0.0 | 0.5 | 100 | 25.39 | 16.15 | 15.25 |
| 0.2 | 0.0 | 0 | 11.10 | 9.84 | 10.03 |
| 0.2 | 0.0 | 30 | 14.87 | 21.64 | 17.05 |
| 0.2 | 0.0 | 100 | 14.92 | **21.69** | 16.97 |
| 0.2 | 0.05 | 0 | 11.15 | 9.82 | 10.03 |
| 0.2 | 0.05 | 30 | 14.89 | 21.62 | 17.00 |
| 0.2 | 0.05 | 100 | 15.03 | 21.65 | 17.03 |
| 0.2 | 0.5 | 0 | 11.08 | 9.82 | 10.03 |
| 0.2 | 0.5 | 30 | 11.10 | 9.80 | 10.03 |
| 0.2 | 0.5 | 100 | 11.10 | 9.82 | 10.03 |
| 1.0 | 0.0 | 0 | 10.00 | 10.01 | 10.00 |
| 1.0 | 0.0 | 30 | 10.00 | 10.00 | 10.00 |
| 1.0 | 0.0 | 100 | 10.05 | 10.00 | 10.00 |
| 1.0 | 0.05 | 0 | 10.00 | 10.01 | 10.00 |
| 1.0 | 0.05 | 30 | 10.00 | 10.00 | 10.00 |
| 1.0 | 0.05 | 100 | 10.00 | 10.00 | 10.00 |
| 1.0 | 0.5 | 0 | 10.00 | 10.01 | 10.00 |
| 1.0 | 0.5 | 30 | 10.00 | 10.01 | 10.00 |
| 1.0 | 0.5 | 100 | 10.00 | 10.01 | 10.00 |

$p$-values. Following standard practice, results with $p < 0.05$ indicate statistically significant performance differences.

Table 29: P-values: FedSOV* vs FedOV, FedGF, FedAvg (paired $t$-test)

| Clients | Dataset | Heterogeneity | FedSOV* vs FedOV | FedSOV* vs FedGF | FedSOV* vs FedAvg |
|---|---|---|---|---|---|
| 5 | MNIST | Extreme | **0.384** | **0.574** | $< 0.05$ |
| 5 | FMNIST | Extreme | **0.088** | $< 0.05$ | $< 0.05$ |
| 5 | SVHN | Extreme | $< 0.05$ | $< 0.05$ | $< 0.05$ |
| 5 | CIFAR-10 | Extreme | **0.059** | **0.069** | **0.125** |
| 5 | CIFAR-100 | Extreme | $< 0.05$ | $< 0.05$ | $< 0.05$ |
| 5 | Tiny-ImageNet | Extreme | $< 0.05$ | **0.103** | $< 0.05$ |
| 10 | MNIST | Extreme | **0.146** | $< 0.05$ | $< 0.05$ |
| 10 | FMNIST | Extreme | $< 0.05$ | $< 0.05$ | $< 0.05$ |
| 10 | SVHN | Extreme | $< 0.05$ | $< 0.05$ | $< 0.05$ |
| 10 | CIFAR-10 | Extreme | $< 0.05$ | $< 0.05$ | $< 0.05$ |
| 10 | CIFAR-100 | Extreme | $< 0.05$ | $< 0.05$ | $< 0.05$ |
| 10 | Tiny-ImageNet | Extreme | $< 0.05$ | $< 0.05$ | $< 0.05$ |
| 20 | MNIST | Extreme | $< 0.05$ | $< 0.05$ | $< 0.05$ |
| 20 | FMNIST | Extreme | $< 0.05$ | $< 0.05$ | $< 0.05$ |
| 20 | SVHN | Extreme | $< 0.05$ | $< 0.05$ | $< 0.05$ |
| 20 | CIFAR-10 | Extreme | $< 0.05$ | $< 0.05$ | $< 0.05$ |
| 20 | CIFAR-100 | Extreme | $< 0.05$ | $< 0.05$ | $< 0.05$ |
| 20 | Tiny-ImageNet | Extreme | $< 0.05$ | $< 0.05$ | $< 0.05$ |
| 5 | MNIST | 0.01 | **0.090** | **0.144** | **0.080** |
| 5 | FMNIST | 0.01 | **0.972** | **0.392** | **0.229** |
| 5 | SVHN | 0.01 | **0.095** | $< 0.05$ | $< 0.05$ |
| 5 | CIFAR-10 | 0.01 | **0.074** | **0.875** | **0.687** |
| 5 | CIFAR-100 | 0.01 | $< 0.05$ | **0.062** | $< 0.05$ |
| 5 | Tiny-ImageNet | 0.01 | $< 0.05$ | **0.850** | **0.074** |
| 10 | MNIST | 0.01 | **0.149** | **0.129** | **0.090** |
| 10 | FMNIST | 0.01 | **0.783** | **0.255** | **0.103** |
| 10 | SVHN | 0.01 | **0.139** | $< 0.05$ | $< 0.05$ |
| 10 | CIFAR-10 | 0.01 | **0.055** | **0.099** | **0.084** |
| 10 | CIFAR-100 | 0.01 | $< 0.05$ | $< 0.05$ | $< 0.05$ |
| 10 | Tiny-ImageNet | 0.01 | $< 0.05$ | $< 0.05$ | $< 0.05$ |
| 20 | MNIST | 0.01 | **0.572** | $< 0.05$ | $< 0.05$ |
| 20 | FMNIST | 0.01 | **0.410** | **0.088** | $< 0.05$ |
| 20 | SVHN | 0.01 | **0.705** | $< 0.05$ | $< 0.05$ |
| 20 | CIFAR-10 | 0.01 | $< 0.05$ | $< 0.05$ | $< 0.05$ |
| 20 | CIFAR-100 | 0.01 | **0.066** | $< 0.05$ | $< 0.05$ |
| 20 | Tiny-ImageNet | 0.01 | $< 0.05$ | $< 0.05$ | $< 0.05$ |
| 5 | MNIST | 0.1 | **0.074** | **0.723** | **0.391** |
| 5 | FMNIST | 0.1 | **0.768** | **0.178** | $< 0.05$ |
| 5 | SVHN | 0.1 | **0.125** | $< 0.05$ | $< 0.05$ |
| 5 | CIFAR-10 | 0.1 | **0.449** | **0.877** | **0.252** |
| 5 | CIFAR-100 | 0.1 | $< 0.05$ | $< 0.05$ | $< 0.05$ |
| 5 | Tiny-ImageNet | 0.1 | $< 0.05$ | **0.162** | **0.111** |
| 10 | MNIST | 0.1 | **0.091** | **0.157** | $< 0.05$ |
| 10 | FMNIST | 0.1 | **0.497** | **0.119** | $< 0.05$ |
| 10 | SVHN | 0.1 | **0.095** | $< 0.05$ | $< 0.05$ |
| 10 | CIFAR-10 | 0.1 | **0.218** | **0.116** | **0.128** |
| 10 | CIFAR-100 | 0.1 | $< 0.05$ | $< 0.05$ | $< 0.05$ |
| 10 | Tiny-ImageNet | 0.1 | $< 0.05$ | **0.092** | **0.176** |
| 20 | MNIST | 0.1 | **0.432** | $< 0.05$ | $< 0.05$ |
| 20 | FMNIST | 0.1 | **0.103** | $< 0.05$ | $< 0.05$ |
| 20 | SVHN | 0.1 | **0.344** | $< 0.05$ | $< 0.05$ |
| 20 | CIFAR-10 | 0.1 | $< 0.05$ | $< 0.05$ | $< 0.05$ |
| 20 | CIFAR-100 | 0.1 | $< 0.05$ | $< 0.05$ | $< 0.05$ |
| 20 | Tiny-ImageNet | 0.1 | $< 0.05$ | $< 0.05$ | $< 0.05$ |

## B.7 SVHN Mode Collapse experiments

n the main results, we observe that parameter-averaging methods collapse dramatically on SVHN. This behavior is consistent with our discussion in Section 3.1, where we mentioned that drift error under parameter averaging may lead to oscillatory dynamics around suboptimal solutions. On SVHN, this manifests as a complete collapse of the learned model. As shown by the confusion matrices, the models degenerate to predicting a single dominant class (class 1) for all inputs, indicating a failure to learn any meaningful decision boundaries as training progresses. A key distinction between SVHN and the other datasets lies in the global

label distribution. SVHN is inherently imbalanced at the dataset level, with certain classes dominating even when aggregating data across all clients. In contrast, the other benchmarks considered are globally balanced, with imbalance arising only at the client level. As a result, on SVHN the global signal itself is biased, and parameter averaging fails to recover a meaningful consensus direction. Consequently, averaging is unable to escape the trivial constant solution, leading to collapse across all parameter-averaging-based methods.

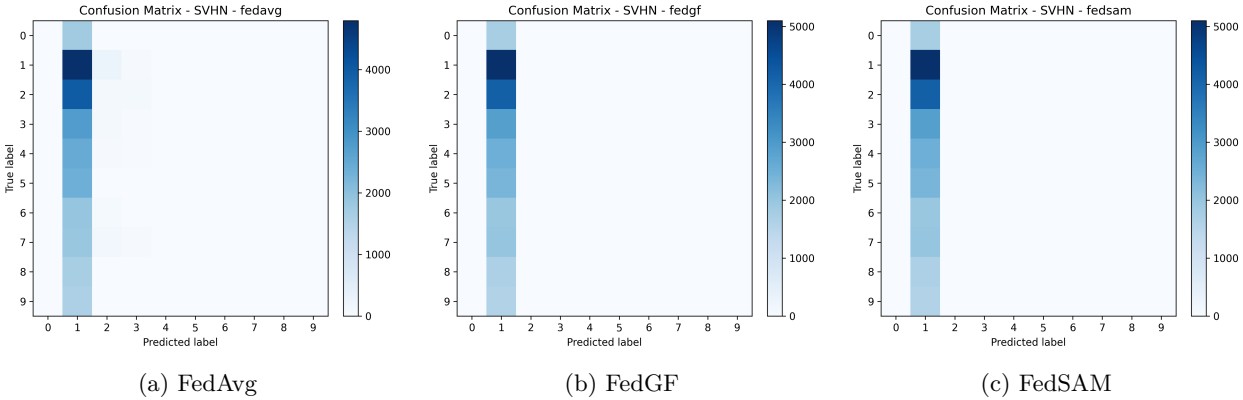

(a) FedAvg      (b) FedGF      (c) FedSAM

Figure 4: SVHN performance under high heterogeneity with 10 clients. Confusion matrices illustrate the collapse due to label imbalance.

Table 30: Class distribution of the SVHN dataset

| Class | 0 | 1 | 2 | 3 | 4 | 5 | 6 | 7 | 8 | 9 |
|---|---|---|---|---|---|---|---|---|---|---|
| # Samples | 4948 | 13861 | 10585 | 8497 | 7458 | 6882 | 5727 | 5595 | 5045 | 4659 |

## C  Additional Discussion

### C.1  Comparison of Ensembles and Parameter Averaging Beyond Data Heterogeneity

Beyond heterogeneity due to label skew, federated learning systems face several practical challenges such as system heterogeneity, model heterogeneity, and continual learning.

**System heterogeneity.** Clients often differ in compute power, communication availability, and participation frequency. Parameter averaging methods like FedAvg require synchronized updates, making them sensitive to such variation. In contrast, ensemble-based methods operate independently: each client updates its own model locally, with no need for synchronization. As a result, ensembles are naturally robust to system heterogeneity.

**Model heterogeneity.** In real-world deployments, clients may have different hardware capabilities or application needs that demand varying model architectures during training. FedAvg assumes identical architectures for parameter alignment, making it incompatible with such settings. Ensemble methods impose no such restriction, as aggregation occurs at the output level, allowing clients to train heterogeneous models without modification during local training.

**Continual learning.** Incorporating continual learning techniques into parameter-averaging frameworks is difficult due to their bilevel optimization structure and global synchronization. Ensemble methods, however, allow each client to update its local model independently using standard continual learning approaches. Clients can fine-tune models on new data and seamlessly update their contribution to the ensemble, enabling modular, scalable, and communication-efficient adaptation over time.

## C.2 Inference Latency in Pruning and Distillation

Inference latency (wall-clock time from input to output) is a critical consideration for the practical adoption of federated ensemble methods, especially in edge and mobile deployments where compute resources are constrained. While accuracy improvements are important, real-world deployment often depends on whether a method can deliver predictions within strict time budgets. In this context, it is valuable to compare the theoretical latency of a pruned ensemble against a distilled monolithic model, as these two represent the primary deployment strategies for the ensemble FL framework

**Note.** To avoid conflicts with previously defined notation, we use nonstandard symbols for several quantities in this section.

We model inference latency in terms of FLOPs, throughput, and overheads. Let $\mathscr{F}$ denote the number of floating point operations (FLOPs) for the baseline monolithic model, $\mu$ the sustained device throughput (FLOPs/s), and $\gamma$ the constant launch and synchronization overhead. The latency of a single monolithic model is

$$\tau_{\mathrm{mono}} \approx \gamma + \frac{\mathscr{F}}{\mu}.$$

For an ensemble of $C$ (number of clients) such models running across $n \leq C$ concurrent execution contexts, the latency is

$$\tau_{\mathrm{ens,\ full}} \approx \left\lceil \frac{C}{n} \right\rceil \left( \gamma + \frac{\mathscr{F}}{\mu} \right).$$

where $\lceil \cdot \rceil$ denotes the ceiling operator, i.e., rounding up to the nearest integer.

We now introduce pruning. If a fraction $\mathcal{P}$ of FLOPs is removed, then under the assumption of *linear FLOP reduction*, the remaining operations per branch are $(1 - \mathcal{P})\mathscr{F}$. An efficiency factor $\omega \in (0, 1]$ accounts for imperfect sparse kernel utilization ($\alpha = 1$ corresponds to ideal structured pruning). The latency of a single pruned branch is

$$\tau_{\mathrm{branch}} \approx \gamma + \frac{(1 - \mathcal{P})\mathscr{F}}{\omega\mu},$$

and the ensemble latency becomes

$$\tau_{\mathrm{ens,\ pruned}} \approx \left\lceil \frac{C}{n} \right\rceil \left( \gamma + \frac{(1 - \mathcal{P})\mathscr{F}}{\omega\mu} \right).$$

Assuming negligible overhead ($\gamma \approx 0$) and ideal efficiency ($\omega = 1$), the condition for the pruned ensemble to be faster than the monolith is

$$\tau_{\mathrm{ens,\ pruned}} \leq \tau_{\mathrm{mono}} \quad \Longleftrightarrow \quad (1 - \mathcal{P}) \leq \frac{1}{\lceil C/n \rceil}.$$

This inequality highlights that higher pruning ratios are required when parallel execution is limited. Intuitively, if sufficient parallel contexts are available ($n = C$), then $\lceil C/n \rceil = 1$, and the ensemble is never slower than the single model, since no individual branch is larger than the monolith and all branches can be executed simultaneously. When parallelism is constrained ($n < C$), however, enough FLOPs must be pruned from each branch to compensate for sequential evaluation across groups of branches.

In practice, achieving such speedups requires industry-grade engineering. Our proof-of-concept experiments use standard PyTorch unstructured pruning, which zeros weights but does not reduce actual FLOPs in execution. Without specialized sparse kernels and hardware-optimized inference code, the wall-clock time remains dominated by dense operations. Furthermore, our experimental setup does not exploit inter-branch parallelism, so ensemble latency is higher than distilled latency in practice. The above derivation should therefore be interpreted as a theoretical upper bound, illustrating that under optimal parallelism and pruning conditions, pruned ensembles can match or surpass monolithic models in inference speed, but realizing these gains in deployment requires significant systems-level optimization.

### C.3   Theoretical Training Time Analysis: Pruning vs. Distillation

We first define the baseline training cost for the standard ensemble FL setting without pruning or distillation. Let:

- $C$ = number of clients.
- $E$ = total local training epochs per client.
- $T_{\text{epoch}}$ = wall-clock time to train for one epoch on a given client model.

In one-shot FL, all clients train locally for $E$ epochs in parallel, so the total client-side time budget is:

$$T_{\text{OneShot}} = E \cdot T_{\text{epoch}}.$$

**Pruning.**   For FedSOV, we use iterative pruning with short local training phases between pruning stages, followed by a final fine-tuning phase after the target sparsity is reached. Let:

- $\mathcal{K}$ = number of pruning stages.
- $e_p$ = local epochs per pruning stage.
- $e_f$ = local epochs for final fine-tuning after reaching full pruning ratio.

The total client-side time is then:

$$T_{\text{prune}} = T_{\text{OneShot}} + \mathcal{K} \cdot e_p \cdot T_{\text{epoch}}.$$

**Distillation.**   For distillation, each client first trains locally, then the server performs additional training to transfer the ensemble's knowledge into a single student model. Let:

- $E_{\text{KD}}$ = number of distillation epochs on the server.
- $T_{\text{epoch}}^{\text{server}}$ = time for one server epoch on the student model.

The total training time is:

$$T_{\text{distill}} = T_{\text{OneShot}} + E_{\text{KD}} \cdot T_{\text{epoch}}^{\text{server}}.$$

**Comparative Analysis.**   The difference in total training time between pruning and distillation is governed by the additional overhead terms $E_{\text{KD}} \cdot T_{\text{epoch}}^{\text{server}}$ and $\mathcal{K} \cdot e_p \cdot T_{\text{epoch}}$, respectively. Which approach is more time-efficient therefore depends on the relative magnitude of these quantities. In practice, however, server-side distillation typically requires a sufficiently large number of distillation epochs to integrate knowledge from all $C$ client models into a single student network. As a result, the server-side cost $E_{\text{KD}} \cdot T_{\text{epoch}}^{\text{server}}$ is often substantially larger than the cumulative pruning overhead $\mathcal{K} \cdot e_p \cdot T_{\text{epoch}}$ incurred by FedSOV. Nevertheless, this comparison is context-dependent and may vary with model size, server compute capacity, and the complexity of the client ensemble.

