# OpenReview forum: "Scalable Ensemble Federated Learning with Enhanced Open-Set Recognition"
_TMLR — Accepted by TMLR_

### Review · Reviewer_KkP7 · 2026-02-25

**Summary Of Contributions:**

The paper addresses the problem of optimization drift in FL under heterogeneous data distributions. Usually, algorithms such as FedAVG tend to update a single model, making consensus difficult to maintain.
An alternative to this classic approach is the use of Enseble with the abstention approach, which avoids parameter consensus by preserving client specialization.
However, this solution has two main challenges: its performance depends on the OSR task, and the model size grows linearly in the number of clients, limiting scalability.
To solve this problem, the authors introduce a new algorithm called FedSOV that improves the OSR mechanism while being more scalable thanks to pruning. Additionally, the paper provides a comparison between parameter averaging and ensemble aggregation and supports the proposed methodology with empirical evaluations across multiple datasets and different settings.

**Audience:**

Yes

**Audience Explanation:**

I believe that the solution introduced in the paper can be useful in many scenarios where we have to learn models with distributed data. A few changes are still needed, but I am confident that people will be interested in this work.

**Broader Impact Concerns:**

I do not have any ethical concerns about this work.

**Claims And Evidence:**

No

**Claims Explanation:**

The paper introduces an interesting approach trying to solve the scalability limits of solutions such as FedOV and proposing a method that can be used in the ensemble with an abstention distributed learning scenario. They evaluated their method using multiple datasets, and I appreciated the fact that they also tested on 50 and 100 clients to evaluate the scalability.
I believe that the paper still needs some things to be ready, in particular in the validation and evaluation.

1. My first concern is about the experimental setup presented in Section 5. From my understanding, the authors fixed the learning rate to 0.001. They did not mention the other hyperparameters used during the training, how they were chosen, or whether they are tuned for each individual method considered in the evaluation.
This is particularly important when comparing the different methods because using the same hyperparameters in FedAvg and for the competitors could lead to results that are not really comparable.
Could you please clarify how you performed the hyperparameter tuning and if this was separate for each method?

2. Besides testing the method using the SimpleCNN model, the authors also reported in Table 10 a comparison using a 3 additional larger CNNs. However, this experiment only shows a scenario with 10 clients and a single heterogeneity. This is a bit limited to support their claim. I would encourage authors to evaluate this at least by using more clients. Having more than a single dataset and more than a single distribution would be optimal.

3. An analysis of the total communication cost of your method compared with the others is missing from the paper. Since FedSOV operates in a one-shot setting, while parameter averaging methods require multiple rounds, this could show another benefit of FedSOV, strengthening the paper's contributions.

4. How is the extreme heterogeneity distribution obtained? Could you please explain how the dataset is constructed in this scenario?

5. In the experiments, you used 20 rounds with 5 local epochs on small datasets and 100 with 10 local epochs per round with larger datasets in the case of parameter averaging. The ensemble-based methods instead are trained for 1 round with 10 and 100 local epochs. The choice of this number of local epochs seems arbitrary to me, both in the FedAvg and in the ensemble case.
Usually, the number of local epochs (and sometimes also the number of communication rounds) in FL is treated as a hyperparameter.
In this case, I'm not sure if the comparison is fair. If we count all the local steps, the FedAvg clients would compute 100 and 1000 local epochs against the 10 and 100 local epochs of the ensemble. In theory, the FedAvg would compute more training.
I would try to make this comparison more fair, either by choosing the same number of total local epochs or doing a proper hyperparameter tuning to show how the FedAvg actually compares with the other methods.

6. I would have expected FedSOV to also be the best one when using Dirichlet distributions. Instead, from Tables 4 and 5 I noticed that in some cases, FedOV or FedGF are better than FedSOV. When FedSOV is the best one, in some cases, it is just a small difference. A statistical test would allow us to better understand the differences between the methods.

7. In Section 3.1, Theorem 1 considers the "One-Round Parameter Averaging Optimization Error" introducing a bound.
If I understood correctly, this is considering the error of a single round classic FedAvg.
In general, a single round of communication is not enough to train a model with FedAvg, while it is enough in the case of your FedSOV algorithms. What happens to this bound when multiple rounds of FedAvg are executed? Could you elaborate more on the multiple-round scenarios and explain how these two errors can be compared?

8. A last minor comment, in Theorem 1 and in Equation 3 in Appendix A.1 authors report the parameter averaging drift for the one-round parameter averaging optimization error. In the first case, they add a -1 at the exponent, while in the second case, the -1 is missing. Please check if there is a typo.

**Requested Changes:**

I would appreciate the authors addressing my concerns written above.

---

> ### Author Response · Authors · 2026-03-26
> **Concern 1: Hyperparameter Setting Elaboration**
>
> We thank the reviewer for the thoughtful and constructive feedback. We briefly waited to incorporate feedback across all reviews; however, as the remaining review appears to be delayed, we proceed with responses and revisions based on the currently available feedback. We address the concerns below in sequence.
>
> Reviewer Concern
> > My first concern is about the experimental setup presented in Section 5. From my understanding, the authors fixed the learning rate to 0.001. They did not mention the other hyperparameters used during the training, how they were chosen, or whether they are tuned for each individual method considered in the evaluation. This is particularly important when comparing the different methods because using the same hyperparameters in FedAvg and for the competitors could lead to results that are not really comparable. Could you please clarify how you performed the hyperparameter tuning and if this was separate for each method?
>
> We acknowledge that the original manuscript did not clearly describe the full set of training hyperparameters used in our experiments, and we agree that including these details would improve transparency and reproducibility, and have added the expanded description in Section 5 Experimental Set up.
>
> In our experiments, all hyperparameters were kept identical across methods unless differences were required by the algorithm itself. The learning rate was set to 0.001, following the configuration used in the FedOV implementation. The batch size was set to 128, which was chosen primarily to reduce training time given the large experimental grid spanning multiple datasets, client counts, and heterogeneity settings. The model architecture used in the main experiments is the Simple CNN described in Section 5, and this architecture was kept identical across methods to ensure a fair comparison. As reported in the paper, we additionally performed architecture ablations with larger CNN backbones to verify that the results are not specific to this model choice. Regarding optimization, we used Adam for the ensemble-based methods, following the configuration used in the FedOV paper, and SGD for parameter-averaging methods, following the FedGF paper. This choice is validated through initial pilot experiments, where we observed that parameter-averaging methods tended to converge more reliably with SGD-based optimization, while Adam worked well for the ensemble-based models.
>
> Among the methods evaluated in the main results, only FedGF includes several additional method-specific hyperparameters. The remaining methods evaluated in the main results rely only on standard local training hyperparameters. We therefore performed a small hyperparameter search for FedGF using the Simple CNN model, and we now include the corresponding tuning in the Appendix B.5, Table 27 of the revised manuscript, together with explanations of these parameters.

---

> ### Author Response · Authors · 2026-03-26
> **Concern 2: Model ablation**
>
> > Besides testing the method using the SimpleCNN model, the authors also reported in Table 10 a comparison using a 3 additional larger CNNs. However, this experiment only shows a scenario with 10 clients and a single heterogeneity. This is a bit limited to support their claim. I would encourage authors to evaluate this at least by using more clients. Having more than a single dataset and more than a single distribution would be optimal.
>
> We agree that evaluating larger backbone architectures across additional client scales and heterogeneity settings would further strengthen the empirical validation.
>
> In the original submission, the larger architecture experiments were limited because training these models across many federated configurations (datasets, heterogeneity regimes, and client counts) was computationally expensive. As a result, the initial ablation focused on a representative setting to verify that the observed gains were not specific to the Simple CNN backbone.
>
> Following the reviewer’s suggestion, we have expanded these experiments in the revised version. In particular, we now include additional evaluations with 20 clients and multiple heterogeneity settings in Section 5 (Table 10, 11, 12), rather than only the extreme heterogeneity scenario. We have also added the FedAvg baseline to these model ablations to provide a clearer comparison of how parameter averaging scales relative to the proposed method. These additional experiments confirm that the relative performance advantage of FedSOV over parameter averaging and prior ensemble methods remains consistent as model architecture changes.  A few additional ablation results on model architecture are added in Appendix B.2.

---

> ### Author Response · Authors · 2026-03-26
> **Concern 3  : Communication cost discussion**
>
> > An analysis of the total communication cost of your method compared with the others is missing from the paper. Since FedSOV operates in a one-shot setting, while parameter averaging methods require multiple rounds, this could show another benefit of FedSOV, strengthening the paper's contributions.
>
> We agree that an explicit communication cost analysis would strengthen the presentation. FedSOV operates in a one-shot setting requiring a single communication round, whereas parameter averaging methods require 20–100 rounds depending on the dataset (Section 5). Since the transmitted model sizes are known, we have included a quantitative comparison of the total communication volume across methods (Table 6) in the revised version. We thank the reviewer for this suggestion.

---

> ### Author Response · Authors · 2026-03-26
> **Concern 4: Extreme Heterogeneity explanation**
>
> >How is the extreme heterogeneity distribution obtained? Could you please explain how the dataset is constructed in this scenario?
>
> To simulate maximal label distribution skew, we construct a label-disjoint partition. Let $N$ denote the number of classes and $C$ the number of clients.
>
> - If $C \le N$, the $N$ classes are evenly partitioned across clients such that each client receives exactly $N/C$ exclusive classes. No class is shared between clients.
>
> - If $C > N$, each client is assigned data from exactly one class. When multiple clients correspond to the same class, the samples of that class are evenly divided among those clients.
>
> Formally, when $C \le N$, the client label supports satisfy
>
> $$
> \mathcal{Y}_i \cap \mathcal{Y}_j = \emptyset \quad \text{for all } i \neq j,
> $$
>
> and when $C > N$, each client observes samples from a single class only.
>
> This construction induces maximal distribution heterogeneity and significantly amplifies parameter drift in consensus-based optimization methods. We have expanded the explanation in Section 5 (page 11) in the revised version.

---

> ### Author Response · Authors · 2026-03-26
> **Concern 5: Fairness of local epochs**
>
> > In the experiments, you used 20 rounds with 5 local epochs on small datasets and 100 with 10 local epochs per round with larger datasets in the case of parameter averaging. The ensemble-based methods instead are trained for 1 round with 10 and 100 local epochs. The choice of this number of local epochs seems arbitrary to me, both in the FedAvg and in the ensemble case. Usually, the number of local epochs (and sometimes also the number of communication rounds) in FL is treated as a hyperparameter. In this case, I'm not sure if the comparison is fair. If we count all the local steps, the FedAvg clients would compute 100 and 1000 local epochs against the 10 and 100 local epochs of the ensemble. In theory, the FedAvg would compute more training. I would try to make this comparison more fair, either by choosing the same number of total local epochs or doing a proper hyperparameter tuning to show how the FedAvg actually compares with the other methods.
>
>
> The difference in total compute between the two paradigms reflects a structural distinction between parameter-averaging and ensemble-based approaches. Parameter-averaging methods rely on iterative global synchronization to progressively reduce consensus error under heterogeneous data distributions. As discussed in Section 3, aggregation drift accumulates with local updates and client dissimilarity, which typically necessitates multiple communication rounds for the model to approach convergence.
>
> Nevertheless, we agree that an additional comparison with a matched compute budget can help clarify the differences between the two approaches. In accordance with this suggestion, we conducted additional experiments in which parameter-averaging methods were trained with 20 communication rounds and 5 local epochs, resulting in a similar total number of local update steps as the 1-round, 100-epoch training used for ensemble-based methods.
>
> These experiments are included in the CIFAR-10 architecture ablation table (Table 10,11,12), where all evaluated architectures were trained under this configuration. To facilitate faster convergence under this reduced-round setting, we used a learning rate of 0.01 for the FedAvg. The results show that FedSOV continues to outperform the parameter-averaging baselines under this matched-compute configuration. This additional experiment confirms that the performance advantage of FedSOV does not depend on unequal compute budgets. We also adopted this configuration in the FedGF hyperparameter ablation experiments in Table 27.

---

> ### Author Response · Authors · 2026-03-26
> **Concern 6: Dirichlet result comparison**
>
> > I would have expected FedSOV to also be the best one when using Dirichlet distributions. Instead, from Tables 4 and 5 I noticed that in some cases, FedOV or FedGF are better than FedSOV. When FedSOV is the best one, in some cases, it is just a small difference. A statistical test would allow us to better understand the differences between the methods.
>
> It is true that the relative differences between methods can be smaller under certain Dirichlet settings. This behavior is expected because moderate heterogeneity reduces the need for strong client specialization. Under Dirichlet-based heterogeneity (particularly for larger values of α), the distribution skew is milder compared to the extreme label-disjoint setting. In such cases, the induced parameter drift is reduced, and consensus-based methods suffer less from optimization misalignment. This observation is consistent with our theoretical analysis in Section 3, where the drift term scales with the level of heterogeneity. Since FedSOV is designed to mitigate the effects of severe heterogeneity through improved open-set handling, its relative advantage becomes more pronounced as heterogeneity increases.
>
> Following the reviewer’s suggestion, we have additionally conducted paired statistical significance tests across random seeds, and we now report the corresponding p-values in an additional table in Appendix B.6 (Table 28) to better quantify the observed differences.

---

> ### Author Response · Authors · 2026-03-26
> **Concern 7: Theoretical Comparison**
>
> > In Section 3.1, Theorem 1 considers the "One-Round Parameter Averaging Optimization Error" introducing a bound. If I understood correctly, this is considering the error of a single round classic FedAvg. In general, a single round of communication is not enough to train a model with FedAvg, while it is enough in the case of your FedSOV algorithms. What happens to this bound when multiple rounds of FedAvg are executed? Could you elaborate more on the multiple-round scenarios and explain how these two errors can be compared?
>
> We thank the reviewer for the question. The goal of Theorem 1 is not to analyze the full convergence behavior of multi-round FedAvg, which has already been extensively studied in the literature [1], [2], [3]. Instead, the theorem isolates the aggregation error introduced by parameter averaging under heterogeneous client objectives, allowing us to compare the two aggregation paradigms.
> Specifically, the result characterizes the per-round drift that arises when multiple local updates are performed before averaging. In multi-round FedAvg, this drift does not disappear; it occurs at every round and must be compensated either through additional communication rounds or by reducing the number of local updates. The purpose of the theorem is therefore to highlight that parameter averaging introduces an inherent aggregation error that grows with both the number of local steps and the degree of client heterogeneity.
>
> In contrast, the ensemble-with-abstention framework does not suffer from this optimization drift. Instead, its performance depends primarily on the accuracy of the expert-selection mechanism (OSR). The theoretical comparison therefore highlights that the two paradigms are sensitive to heterogeneity in fundamentally different ways.
>
> [1] X. Li, K. Huang, W. Yang, S. Wang, and Z. Zhang, On the Convergence of FedAvg on Non-IID Data, International Conference on Learning Representations (ICLR), 2020.
>
> [2] T. Li, A. K. Sahu, A. Talwalkar, and V. Smith, Federated Optimization in Heterogeneous Networks, MLSys, 2020.
>
> [3] S. P. Karimireddy, S. Kale, M. Mohri, S. Reddi, S. Stich, and A. T. Suresh, SCAFFOLD: Stochastic Controlled Averaging for Federated Learning, ICML, 2020.

---

> ### Author Response · Authors · 2026-03-26
> **Concern  8: Typo**
>
> >A last minor comment, in Theorem 1 and in Equation 3 in Appendix A.1 authors report the parameter averaging drift for the one-round parameter averaging optimization error. In the first case, they add a -1 at the exponent, while in the second case, the -1 is missing. Please check if there is a typo.
>
> We thank the reviewer for carefully identifying this inconsistency. This is a typo, and the correct exponent should be T−1, which is consistent with the full derivation presented in Appendix A. We have corrected the typo in the revised manuscript. We appreciate the reviewer’s careful reading.

---

### Review · Reviewer_CCCS · 2026-03-09

**Summary Of Contributions:**

This paper seeks an alternative to parameter averaging for global model updates in federated learning. The alternative presented is FedSOV, an ensemble based, one shot federated learning method that the authors argue performs better than parameter averaging under extreme cases of client data heterogeneity. The difficulty of scaling ensemble methods is addressed by pruning.

**Additional Comments:**

While I think investigation of alternatives to parameter averaging is a relevant and interesting research direction, I am concerned about the novelty and the experimental justifications in this work. I did find the method descriptions and theoretical justifications unclear, so it may be the case that some clarification in this area would alleviate some of these concerns.

**Audience:**

No

**Audience Explanation:**

I have two major concerns with the papers findings:

1. I am concerned about the novelty of the contributions. "Smoothed shuffled negatives" was presented as a novel improvement but not explained at all and has no theoretical or empirical justification in this work. The only other novel contribution seems to be model pruning to address the high cost of ensembling. The lottery ticket hypothesis, published in 2019 seems to be the most sota pruning method considered and from the discussion presented in the paper, it does not appear that any attempt was made to improve upon this strategy or further tailor it for FL purposes.

2. Given the evolution of neural networks towards increasingly large parameter counts, the fact that the bulk of the experiments were conducted on a 2-layer CNN significantly lowers the relevancy of this work. There are some examples presented using larger (but still very small) models and it appears that the results on these larger models underperform the 2-layer CNN example. This suggests to me that the method may not scale well, but additional experiments would be required to make that conclusion including a FedAvg baseline so we can see if ensembling truly does improve over parameter averaging with larger scale models.

The idea of pruning to reduce the number of parameters required is interesting but it was not well explored. The pruning strategies need to be better defined and the results more clearly articulated. Also, pruning a 2-layer CNN is not particularly useful, larger models need to be considered.

**Claims And Evidence:**

No

**Claims Explanation:**

The problem setting is clearly defined and there is merit to investigating ensembling as a means to reduce client drift in federated updates and to developing one-shot solutions that reduce communication in FL. However, the method formulation and justification are very unclear.

Some specific examples include:
* OSR is central to the method formulation and referenced extensively but never formally defined. Section 4.1 devotes three paragraphs to defining the OSR task without showing an objective function or explaining terms such as "random cut-and-paste negatives"  and "smoothed shuffled negatives". Smoothed shuffled negatives in particular should be clarified since it is presented as a novel improvement introduced by the authors
*  Theorems 1 and 2 and Preposition 1 need to be better formulated and justified, moving some of the assumptions from the appendix into the main paper might help with this. One specific example is gradient dis-similarity $B_c$, how is this defined? How is it bounded? Given the claims you make, this does need to be better justified in the main paper.
*  An algorithm block could be very helpful since the entire method is a bit unclear, I would particularly appreciate some clarification on how ensembling is done once the client models are obtained
* The main experiments are conducted on a 2-layer CNN calling into question the broad applicability of this method. Since model pruning is part of the proposed method, I would also be very curious to see how larger models react to pruning.

**Requested Changes:**

* The FedSOV method, including the theoretical justifications and OSR need to be more clearly defined. An algorithm block would help a lot
* The use of "smoothed shuffled negatives" should be explained and justified in some way
* The method needs to be justified using larger models, a 2-layer CNN is a toy example, particularly since model pruning is the main novelty of this method.

---

> ### Author Response · Authors · 2026-03-26
> **Response to OSR Formulation and Negative Sample Generation Clarity**
>
> We thank the reviewer for their detailed engagement with our work and for the constructive feedback. We briefly waited to incorporate feedback across all reviews; however, as the remaining review appears to be delayed, we proceed with responses and revisions based on the currently available feedback. We organize our response into three parts addressing (1) clarity of the formulation, (2) model architecture and experimental scope, and (3) novelty of the contributions.
>
> Reviewer Concern:
> >OSR is central to the method formulation and referenced extensively but never formally defined. Section 4.1 devotes three paragraphs to defining the OSR task without showing an objective function or explaining terms such as 'random cut-and-paste negatives' and 'smoothed shuffled negatives'. Smoothed shuffled negatives in particular should be clarified since it is presented as a novel improvement introduced by the authors.
>
> We appreciate the reviewer’s suggestion. In the current version we focused on the benefit of Open Set Recognition (OSR) for FL and may have assumed too much familiarity with the task. We agree that adding a formal objective and clearer definitions would improve readability, and we have included these clarifications (section 4.1 page 9) in the revised version.
> In section 4.1 we have added a clearer explanation. The OSR objective trains a model not only to classify samples from known classes during training, but also to reject samples that do not belong to any known class at test time. In the context of federated learning, this property enables models to abstain on unfamiliar inputs, which allows the ensemble to route data to the most appropriate client model rather than forcing consensus through parameter averaging.
>
> We also agree that the description of the negative samples can be expanded and have added the following explanation in section 4.1.
>
> In FedOV, negatives are generated using random cut-and-paste augmentations, where patches are extracted from an image and relocated or rotated. While this creates visually corrupted images, it also introduces artificial discontinuities that do not appear in natural unknown samples. As a result, the model may rely on these artifacts as shortcut cues when learning to reject negatives.
> Our smoothed shuffled negatives are designed to mitigate this issue. Instead of introducing sharp discontinuities, we generate negatives by shuffling image regions and applying smoothing operations to remove boundary artifacts. This preserves low-level statistics while removing semantic structure, producing negatives that better resemble realistic unknown inputs.
>
> Regarding the reviewer’s comment that the contribution lacks theoretical justification and empirical justification
> > "Smoothed shuffled negatives" was presented as a novel improvement but not explained at all and has no theoretical or empirical justification in this work.
>
> The empirical benefit of this design can be observed in the comparison between FedOV and FedSOV in Table 2, 3 and 4, where improvements in ensemble performance correlate with improved OSR behavior. More specifically, Table 22 reports AUROC scores measuring open-set detection performance, which show consistent gains with the proposed negative construction. The comparison between the full ensemble and FedOV serves as a direct ablation of the effect on OSR of the smoothed shuffled negatives. We agree that making this connection clearer in the paper would strengthen the presentation, and we have revised the manuscript to highlight this ablation more explicitly in Appendix B.3. If the reviewer feels that additional controlled ablations would further clarify the effect, we would be happy to consider them.
>
> For the theoretical justification concern, we note that Section 4.1 provides a qualitative analysis of the failure mode of cut-and-paste negatives and the motivation for the proposed construction. In particular, the goal is to construct negative samples that retain high similarity to known classes at the level of low-level statistics, while exhibiting low class membership and avoiding shortcut cues. These properties can be reasoned about from a representation learning perspective and serve as guiding principles for negative sample design.
>
> However, while these properties may be theoretically motivated, the exact construction that optimally realizes them in the input space is not uniquely defined. In practice, such constructions are guided by these principles and validated empirically. If the reviewer is instead referring to a more formal analysis of these properties or their effect on learned representations, we would be happy to clarify this point and include additional analysis in the appendix of the revised version. We would appreciate further guidance on the specific type of theoretical justification the reviewer has in mind.

---

> > ### Author Response · Authors · 2026-03-26
> > **Response to Theoretical Assumptions**
> >
> > Reviewer Concern
> > >Theorems 1 and 2 and Preposition 1 need to be better formulated and justified, moving some of the assumptions from the appendix into the main paper might help with this. One specific example is gradient dis-similarity, how is this defined? How is it bounded? Given the claims you make, this does need to be better justified in the main paper.
> >
> > We thank the reviewer for the suggestion. The assumptions used in Theorem 1 follow standard formulations widely used in theoretical analyses of federated learning under data heterogeneity, and our intent was to adopt these commonly used assumptions rather than introduce new ones. In particular, the bounded gradient dissimilarity assumption is frequently used in FL literature to characterize the degree of client heterogeneity.
> >
> > Formally, the assumption states that the difference between the local client gradient and the global gradient is bounded by a client-specific constant:
> >
> > $$
> > \|\nabla L_c(\theta) - \nabla L(\theta)\| \le B_c
> > $$
> >
> > where $L_c(\theta)$ denotes the local loss for client $c$, $L(\theta)$ denotes the global objective, and $B_c$ captures the level of heterogeneity of client $c$. Intuitively, this assumption states that while client gradients may differ due to heterogeneous data distributions, their deviation from the global gradient is bounded.
> >
> > Since this assumption is commonly adopted in theoretical analyses of federated learning [2], [3], our goal in Theorem 1 was to analyze the aggregation behavior under standard heterogeneity assumptions rather than introduce new modeling assumptions. That said, we agree with the reviewer that moving the statement of these assumptions from the appendix to the main text in section 3.1 would improve clarity, and we have revised the manuscript accordingly.
> >
> > For Theorem 2, the full set of assumptions and supporting definitions is provided in the appendix, together with additional intuition regarding their role in the analysis. Because the result requires several auxiliary definitions and technical conditions, we place the detailed assumptions and definitions in the appendix to preserve the readability of the main text. If the reviewer prefers that the complete assumptions be moved to the main paper, we would be happy to do so.
> >
> > [1] X. Li, K. Huang, W. Yang, S. Wang, and Z. Zhang, On the Convergence of FedAvg on Non-IID Data, International Conference on Learning Representations (ICLR), 2020.
> >
> > [2] T. Li, A. K. Sahu, A. Talwalkar, and V. Smith, Federated Optimization in Heterogeneous Networks, MLSys, 2020.
> >
> > [3] S. P. Karimireddy, S. Kale, M. Mohri, S. Reddi, S. Stich, and A. T. Suresh, SCAFFOLD: Stochastic Controlled Averaging for Federated Learning, ICML, 2020.

---

> > ### Author Response · Authors · 2026-03-26
> > **Pseudo Algorithm Box and Ensembling Clarification**
> >
> > Reviewer Concern
> > >An algorithm block could be very helpful since the entire method is a bit unclear, I would particularly appreciate some clarification on how ensembling is done once the client models are obtained.
> >
> > We thank the reviewer for this suggestion. A pseudocode description of the FedSOV procedure is currently included in Appendix D, and we agree that moving this algorithm block to the main paper (Page 8) would make the workflow clearer. We have revised the manuscript accordingly.
> >
> > To briefly clarify the ensembling procedure: each client model is trained using the known classes and the synthetic negatives mentioned in figure 2 stagewise. The training includes an additional unknown-class logit, which is used to classify the synthetic negatives. At inference time, this logit provides a confidence score indicating how likely a sample is to belong to an unknown class for that particular client model. These scores are used to determine which client models are confident on a given input. Following the FedOV framework, the top-k models with the lowest unknown-class confidence (i.e., highest confidence in known classes) are selected, and their predictions are aggregated to produce the final ensemble output.

---

> ### Author Response · Authors · 2026-03-26
> **Model Architecture and Scalability Concerns**
>
> Reviewer Concern
> >The main experiments are conducted on a 2-layer CNN calling into question the broad applicability of this method. Since model pruning is part of the proposed method, I would also be very curious to see how larger models react to pruning.
>
> The Simple CNN architecture was used for the majority of experiments primarily because the evaluation requires a large experimental grid across multiple datasets, heterogeneity settings, and client counts. Simple CNN-style architectures with a small number of convolutional and fully connected layers are commonly used in federated learning studies because they train quickly while achieving strong performance on standard FL benchmarks such as CIFAR-10, CIFAR-100, SVHN, MNIST and Fashion-MNIST which are not very large datasets.
> A further practical consideration is that ensemble-based federated learning methods scale model size linearly with the number of clients, since each client contributes a model to the ensemble. For example, a 20-client system results in a 20× increase in model size, while a 100-client system results in a 100× increase. As shown in Table 5, even with relatively small base models such as Simple CNN, the resulting ensemble can become very large.
>
> Nevertheless, we agree that evaluating larger architectures is important to assess whether the proposed method is architecture-agnostic, and that the original model ablation was limited in this regard. Following the reviewer’s suggestion, we have expanded the model ablation experiments at the end of section 5 to include larger architectures under additional client counts and heterogeneity settings, and we have also added FedAvg baselines for comparison. These additional experiments confirm that the relative performance advantage of FedSOV over parameter averaging and prior ensemble methods remains consistent as model architecture changes.  A few additional ablation results on model architecture are added in Appendix B.2.
>
> We note that absolute accuracy for all methods can sometimes decrease when using larger architectures in federated learning benchmarks. This behavior is expected because the available data per client is limited, particularly under heterogeneous partitions. Larger models have higher capacity and can therefore be more prone to overfitting on smaller per-client datasets, whereas simpler architectures such as Simple CNN often generalize more reliably in these settings.

---

> ### Author Response · Authors · 2026-03-26
> **Novelty Concerns**
>
> Reviewer Concerns
> > I am concerned about the novelty of the contributions. "Smoothed shuffled negatives" was presented as a novel improvement but not explained at all and has no theoretical or empirical justification in this work. The only other novel contribution seems to be model pruning to address the high cost of ensembling. The lottery ticket hypothesis, published in 2019 seems to be the most sota pruning method considered and from the discussion presented in the paper, it does not appear that any attempt was made to improve upon this strategy or further tailor it for FL purposes.
>
> We appreciate the reviewer raising this point and we would like to clarify the intended scope of our contributions since we may not have communicated it clearly. The novelty of the paper is not limited to a particular OSR augmentation or a specific pruning technique, but rather to the broader insight regarding alternative aggregation paradigms for heterogeneous federated learning.
> The dominant paradigm in federated learning is parameter averaging, which most existing methods build upon with additional optimization or regularization techniques. An alternative paradigm based on ensembles with abstention has been explored primarily in the context of communication-efficient one-shot federated learning, but it has generally been viewed as a compromise due to two key challenges:
>
> (1) performance degradation under heterogeneous data distributions, and
>
> (2) lack of scalability due to the linear growth of ensemble size with the number of clients.
>
> Our work aims to investigate whether this alternative paradigm can in fact become competitive with SOTA parameter averaging methods when these two limitations are addressed.
>
> The first contribution of this paper is a theoretical analysis comparing parameter averaging and ensemble-based aggregation under heterogeneous client objectives. We show that while ensemble methods depend on the effectiveness of the Open Set Recognition (OSR) mechanism used to route inputs to client models, improving this component fundamentally removes their primary limitation. Under this condition, ensemble-based aggregation is inherently more robust to client heterogeneity than parameter averaging. To empirically support this analysis, we introduce an improved OSR construction that reduces shortcut cues present in previous approaches. The purpose of this design is not to claim that this specific augmentation is the final or optimal OSR solution, but rather to demonstrate that improving OSR directly improves the effectiveness of the ensemble-based paradigm under heterogeneity.
>
> The second contribution addresses the well-known scalability limitation of ensemble-based federated learning. Even if ensemble methods achieve strong performance under heterogeneity, they remain impractical unless the resulting model size can be controlled. Existing approaches typically rely on distillation, which requires access to server-side public data with sufficient diversity. Our work shows that pruning can serve as an effective alternative to distillation for compressing ensemble models, enabling scalability without requiring additional data. The key insight here is not a new pruning algorithm itself, but the finding that appropriately applied pruning can recover a compact model that preserves the benefits of the ensemble aggregation strategy. Our goal was to isolate whether pruning can mitigate ensemble growth rather than introduce an entirely new pruning algorithm. For this reason, we adopt a well-established pruning approach from the literature in order to isolate the effect of pruning on ensemble scalability.
>
> We hope this clarification better communicates the intended contributions of the paper.

---

### Review · Reviewer_LeAe · 2026-04-08

**Summary Of Contributions:**

## Summary of Contributions

FedSOV extends FedOV (Diao et al., ICLR 2023) with two additions:
-  **smoothed shuffled negatives** for improved OSR: image patches are shuffled and boundaries smoothed, preserving class-agnostic features while destroying spatial semantics; and
-  **lottery-ticket pruning** to reduce ensemble size to a single model's parameter count.

A theoretical framework (Theorems 1-2, Proposition 1) compares parameter averaging drift vs. ensemble OSR mismatch. Experiments span 6 vision datasets, 3 heterogeneity regimes, and 5-100 clients.

**Strengths**:
- Well-motivated framing of ensemble-with-abstention;
- principled negative generation with clear shortcut-cue argument;
- pruning addresses a real scalability bottleneck;
- comprehensive experimental grid with honest appendix (LR ablations, p-values, SVHN collapse analysis).

**Weaknesses**:
- Correctable proof error in Theorem 1 (Eq. 11);
- only Simple CNN (~150K params) in main experiments;
- FedSOV underperforms FedOV in several Dirichlet 0.01/0.1 settings;
- headline claims overstate generality;
- missing comparison with FENS (NeurIPS 2024) and Co-Boosting (ICLR 2024);
- unstructured pruning caveat buried in appendix.

Core claims are supported. The paper makes meaningful contributions (smoothed shuffled negatives, pruning for ensemble scalability, theoretical framing) but needs a proof fix, qualified claims, missing baselines (FENS, Co-Boosting), and transparency about pruning limitations. With these corrections, it would be a solid contribution to ensemble FL.

**Audience:**

Yes

**Audience Explanation:**

Ensemble-with-abstention vs. parameter averaging is a valuable framing for FL. Smoothed shuffled negatives are applicable beyond FL. Pruning vs. distillation comparison (Tables 9, 23-26) provides actionable practitioner guidance.

**Broader Impact Concerns:**

No significant concerns. One-shot communication may improve privacy by reducing model upload frequency. The unstructured pruning presentation issue is addressed in C4.

**Claims And Evidence:**

Yes

**Claims Explanation:**

**Theorem 1 proof error** (Eq. 11, p.20): Replaces average-of-norms with norm-of-average inside an upper bound, which is invalid by the triangle inequality. The corrected bound should have exponent $T$ instead of $T{-}1$. Qualitative conclusion (exponential drift) unchanged; error is correctable.

**"18.81% gain over FedOV"**: Supported at 10+ clients under extreme heterogeneity. Overstated as a general claim, FedSOV underperforms FedOV in several settings (e.g., MNIST 10-client Dir 0.01: 54.15 vs 61.33; FMNIST 20-client Dir 0.01: 59.52 vs 62.83). Many 5-client comparisons not significant (Table 28: p=0.384, 0.972, 0.074 across settings).

**"92.43% over FedGF"**: Reflects SVHN-specific mode collapse (class 1 has 3x more samples than class 9), not general superiority. The underlying point (ensembles are more robust under extreme heterogeneity) is well-supported across other datasets.

**Pruning as a practical solution**: Parameter count reduction supported (Tables 5, 9). But pruning is unstructured (weight zeroing, no FLOPs reduction), acknowledged in Appendix C.2, but not in the main text.

**Requested Changes:**

## Critical

**C1. Fix Theorem 1 proof.** Track individual client deviations instead of averaged deviation. Corrected bound has exponent $T$ not $T{-}1$.

**C2. Compare with FENS (NeurIPS 2024) and cite Co-Boosting (ICLR 2024).** FENS reports ~68% on CIFAR-10 under comparable settings vs. FedSOV's ~50%. Co-Boosting reports +8.83% over baselines. Both are ensemble OFL methods, and without them, state-of-the-art competitiveness is unclear.

**C3. Qualify headline claims.** Report gains per heterogeneity regime. Acknowledge settings where FedSOV underperforms FedOV. Reframe the 92.43% SVHN number.

**C4. Move the unstructured pruning caveat to the main text.** Readers should know upfront that weight zeroing does not reduce FLOPs.

**C5. Discuss statistical significance in the main results.** Many 5-client comparisons are not significant -- this should not be relegated to the appendix.

---

> ### Author Response · Authors · 2026-04-20
> **Response to Reviewer Concerns**
>
> We thank the reviewer for their careful and thorough evaluation of our work, as well as for their constructive and insightful suggestions. We address the requested changes below.
>
> **Requested Change 1:**
>
> Thank you for pointing out the issue in Equation 11 of Theorem 1. We have revised the proof to correctly track the average of client deviations (i.e., the average of norms) rather than the norm of the averaged deviation, and subsequently bound the deviation of the aggregated model using the triangle inequality, which resolves the mathematical inconsistency in the original argument.
> It is important to note that the revised bound, expressed in terms of $T$, does not explicitly capture the exact equivalence at the first step and is therefore looser in this respect. At the first local step, the averaged update is exactly equivalent to the centralized gradient step due to linearity, i.e., $\frac{1}{C}\sum_{c=1}^C \nabla L_c(\theta^0) = \nabla L(\theta^0)$, implying $\bar{\theta}^{(1)} = \tilde{\theta}^{(1)}$ and hence zero deviation at $t=1$. However, this cancellation holds only at initialization; for subsequent steps, local models evolve at different points and this linearity no longer applies, so it cannot be incorporated into a general recursive bound. Accordingly, we have corrected the proof and updated the final bound to reflect a $(1+\eta\beta)^T$ dependence, while moving the discussion of the exact cancellation at $t=1$ to a remark as it reflects a structural property of the first step rather than a general bound.
>
> **Requested Change 2:**
>
> We thank the reviewer for pointing out these relevant works. We have added results and discussion for FENS (NeurIPS 2024) and Co-Boosting (ICLR 2024) in the extended comparisons in the Appendix B.1 (Pages 30 and 31).
>
> Our results indicate that FedSOV outperforms these methods in the evaluated settings, particularly under more extreme heterogeneous regimes where their performance degrades. Co-Boosting involves a large number of method-specific hyperparameters (e.g., synthetic loss weights, generator settings, and distillation parameters). Given this complexity, we adopt the default configurations provided in the official implementations rather than performing extensive re-tuning for our setting. Accordingly, we place greater emphasis on comparison with FENS (Table 14), as it allows for a more controlled evaluation. The results show a clear trend: under higher heterogeneity, FENS underperforms relative to both FedSOV and FedOV, while its performance converges toward them as the degree of homogeneity increases. We provide a more detailed discussion of this behavior in Appendix B.1 (Page 31).
>
> We note that these methods are not evaluated under identical settings to those reported in their original papers. Both FENS and Co-Boosting are typically studied under different experimental configurations, including model architectures, and overall training budgets, often operating in higher-compute regimes. To ensure a fair comparison, we evaluate all methods within a unified setup, using the same model architectures and compute budgets as other baselines in our study.
>
> **Requested Change 3:**
>
> We thank the reviewer for this suggestion. We have removed numerical headline claims from the abstract, as they may be misleading without sufficient context, and instead provide a more nuanced discussion of performance across different heterogeneity regimes in the results section. In particular, we now report results per regime and explicitly acknowledge settings where FedSOV underperforms FedOV (Page 12). We have also mentioned the imbalanced induced effects on the SVHN results in the main text (Page 12).
>
> **Requested Change 4:**
>
> We agree that moving this discussion to the main text (Page 10) provides better context, and have done so in the revised manuscript.
>
> The purpose of this discussion is to clarify that while our current implementation does not realize runtime speedups, unstructured pruning can yield practical acceleration when combined with sparse kernels and hardware support, particularly at high sparsity levels. The limitation is not inherent to unstructured pruning itself, but to the absence of specialized sparse kernels and systems-level support in our current implementation; without such optimizations, zeroed weights do not translate into reduced effective FLOPs at runtime. Accordingly, the derived latency improvements should be interpreted as an upper bound achievable under ideal parallelism and optimized deployment conditions, and we have made this distinction explicit to avoid misinterpretation.

---

> ### Author Response · Authors · 2026-04-20
> **Response to Reviewer Concerns**
>
> **Requested Change 5:**
>
> We thank the reviewer for raising this point. We have added a discussion of statistical significance in the main results section (Page 12). In particular, we clarify that the occasional lack of statistical significance arises from low signal rather than method instability. These cases correspond primarily to low-client, low-heterogeneity regimes on simpler datasets, where performance is saturated and differences between methods are inherently small, leading to high p-values. This observation is consistent with our earlier discussion (Concern 3), as the regimes where FedOV occasionally outperforms FedSOV largely coincide with those exhibiting high statistical variance, indicating that such differences are not reliably distinguishable.

---

### Decision · Action_Editor_QRnZ · 2026-05-19

**Recommendation:** Accept with minor revision

**Additional Comments:**

The theoretical results are relatively simple but provide some  motivation for expert selection and OSR quality improvements I think this can be made more clear. Please revise the discussion of the theoretical analysis to more clearly state its scope and limitations. Some of these have already been covered in the discussions and should be incorporated for example that the authors mentioned in the replies the theory does not imply anything about how to pick the experts should be mentioned. It should be made clearer that the smoothed shuffled negatives are not specifically derived from this.  The first statement in the contributions should also be toned down to reflect the limitations of the analysis.

**Audience:**

Yes

**Audience Explanation:**

FL remains a timely topic and the approach is relevant in some settings of potential interest and as a baseline in 1-shot FL

**Claims And Evidence:**

Yes

**Claims Explanation:**

The paper proposes an extension to FedOV that uses expert selection, smoothed shuffled negatives to improve results and pruning to make the ensemble scalable.
The authors also claim to provide a theoretical analysis to motivate their strategy, this is done through comparison of some simple bounds between ensembling and FedAVG and provides some high level motivation for the method.
The authors provide empirical evidence this improves results and in particular works well under extreme heterogeneity.

---

> ### Author Response · Authors · 2026-06-04
> **Minor Revisions**
>
> We have made the requested changes. On Page 2, we revised the first contribution statement to clarify that the theoretical analysis is intended as a paradigm-level comparison and to avoid any ambiguity that the proposed method is directly derived from the theoretical results.
>
> Additionally, on Page 5 (end of the first paragraph), we explicitly clarify the scope and limitations of the analysis by stating that it should be interpreted as a conceptual comparison between parameter averaging drift and expert-selection error, does not prescribe a specific expert-selection mechanism, and does not derive the proposed smoothed shuffled negative generation strategy. We further clarify that the proposed OSR design is empirically motivated and evaluated experimentally.  We have also slightly shortened the literature review by removing a brief discussion of some theoretical analyses.
>
> Finally, we have de-anonymized the manuscript and associated resources for the camera-ready version, following the instructions provided in the acceptance email.